# Silencing CA1 pyramidal cells output reveals the role of feedback inhibition in hippocampal oscillations

Chinnakkaruppan Adaikkan [1] ✉, Justin Joseph [1,7], Georgios Foustoukos[2,6,7], Jun Wang[3,7], Denis Polygalov [2], Roman Boehringer[2], Steven J. Middleton[2], Arthur J. Y. Huang[2], Li-Huei Tsai [3,4] & Thomas J. McHugh [2,5] ✉

The precise temporal coordination of neural activity is crucial for brain function. In the hippocampus, this precision is reflected in the oscillatory rhythms observed in CA1. While it is known that a balance between excitatory and inhibitory activity is necessary to generate and maintain these oscillations, the differential contribution of feedforward and feedback inhibition remains ambiguous. Here we use conditional genetics to chronically silence CA1 pyramidal cell transmission, ablating the ability of these neurons to recruit feedback inhibition in the local circuit, while recording physiological activity in mice. We find that this intervention leads to local pathophysiological events, with ripple amplitude and intrinsic frequency becoming significantly larger and spatially triggered local population spikes locked to the trough of the theta oscillation appearing during movement. These phenotypes demonstrate that feedback inhibition is crucial in maintaining local sparsity of activation and reveal the key role of lateral inhibition in CA1 in shaping circuit function.

Oscillations in the brain reflect the temporal organization of local circuit activity and play key roles in a range of cognitive processes[1,2]. In the CA1 region of the hippocampus, three prominent bands of oscillations dominate the local field potential (LFP); theta (4–12 Hz), gamma (30–100 Hz), and ripples (100–300 Hz), with each thought to contribute distinct, although sometimes overlapping, functions[3–6]. In the awake state, theta increases during locomotion and attention and is necessary for memory function and recall[7–9]. Gamma oscillations are associated with movement velocity, sensory processing, attention, sensory feature binding, and cognition and memory[10–13]. Ripple oscillations, prominent during slow-wave sleep and quiet wakefulness, have been implicated in memory consolidation, storage, and recall[14–18].

The mechanistic understanding of these oscillations suggests that they emerge due to spatially organized, summed current flow within the neural network; thus, they are coupled to underlying fluctuations in synaptic excitation and inhibition[19,20]. However, there are varied mechanisms by which these temporal patterns can emerge within local neural circuits in the hippocampus. For example, oscillations induced by activating mGluR or kainate receptors require GABAergic, but not glutamatergic, synaptic transmission[21,22]. In contrast, the cholinergic receptor agonist carbachol requires glutamatergic neurotransmission, which then engages GABAergic neurotransmission to induce oscillations[23–25]. While providing key insights about the necessity of GABAergic and glutamatergic transmission in specific oscillations, these pharmacological approaches lack anatomical specificity and suffer from redundancy due to the multiple GABA and glutamatergic receptor subunits expressed in the hippocampal circuit. Nonetheless, the canonical interpretation based on both experimental and

[1]Centre for Brain Research, Indian Institute of Science, Bengaluru, Karnataka, India. [2]Laboratory for Circuit and Behavioral Physiology, RIKEN Center for Brain Science, Wakoshi, Saitama, Japan. [3]Department of Brain and Cognitive Sciences, Picower Institute for Learning and Memory, Massachusetts Institute of Technology, Cambridge, MA, USA. [4]Broad Institute of Harvard and Massachusetts Institute of Technology, Cambridge, MA, USA. [5]Department of Life Sciences, Graduate School of Arts and Sciences, The University of Tokyo, Tokyo, Japan. [6]Present address: Department of Fundamental Neurosciences, University of Lausanne, Lausanne, Switzerland. [7]These authors contributed equally: Justin Joseph, Georgios Foustoukos, Jun Wang. ✉e-mail: chinna@iisc.ac.in; thomas.mchugh@riken.jp

computational work suggests that synchronous excitatory input to CA1 from upstream neuronal ensembles engages local circuitry to generate fast oscillations such as gamma and ripples[26–29]. This is consistent with the observation that these ripple oscillations are not directly transferred in this feedforward circuit, and indeed the events themselves have a slower average intrinsic frequency in the upstream CA3 region[6].

Recent optogenetic-mediated interventional studies in mice demonstrated that direct activation of pyramidal neurons in the CA1 is sufficient to drive high-frequency ripple-like oscillations during rest[30], supporting the view that external input to CA1 pyramidal cells, together with local CA1 circuit architecture, underlies ripple generation and maintenance. In contrast, dampening CA1 pyramidal cell excitability by optogenetic activation of inhibitory interneurons suppresses or truncates ripples in mice[30]. Specifically, activation of somatostatin-positive (SST+) interneurons, suggested to primarily supply feedback inhibition (FBI), reduces the amplitude of ripple oscillations[31]. Intriguingly, activation or inhibition of parvalbumin-positive (PV+) interneurons, thought to mainly sub-serve feedforward inhibition (FFI) onto pyramidal neurons, as well as reciprocal inhibition of other interneurons, shortens the duration of ripples[30]. However, many of the numerous subclasses of CA1 interneurons, including SST+ and PV+ neurons, receive multiple excitatory inputs[32,33] and cannot be classified as solely contributing to feedforward to feedback circuits. Thus, while these optogenetic approaches permit exquisite cell-type and temporal specificity, they cannot unambiguously distinguish the unique contributions of these two modes of inhibitory control. Although these data are equivocal, a model has emerged that suggests that activation of pyramidal neurons followed by feedback and reciprocal inhibition (E-I-I) generates and maintains ripple oscillations.

Recent studies both in vitro and in vivo have pointed to the importance of lateral feedback inhibition in controlling the sparsity of pyramidal cell activity in CA1[34–36], arguing for the existence of local circuit motifs that control sparsity and allow flexible scaling of spatial representations. However, it remains unknown how feedback inhibition recruited by local CA1 pyramidal cell spiking specifically contributes to the generation of network activity. Thus, to specifically address the role of feedback inhibition in CA1 oscillations, we employed a genetic approach to chronically silence CA1 pyramidal cell transmission. Given the lack of appreciable recurrent excitatory connections in CA1[33] (but also see refs. [37,38]) the local circuit effects of this intervention is a loss of the ability of CA1 pyramidal cells to recruit FBI, while leaving feedforward inhibition and excitation intact. This decoupling of FBI from local activity allows us to address its role in shaping population activity and oscillations in CA1. Our data reveal that chronic silencing of CA1 pyramidal cell transmission robustly reduces feedback inhibition, leading to increases in the power, duration, and intrinsic frequency of ripple oscillations. Moreover, during movement normal place cell activity is occluded by the appearance of local population spikes, phase locked to theta, that exhibit spatially triggered activity.

## Results

### Selective loss of neurotransmission in CA1 pyramidal neurons leads to a reduction in feedback inhibition

To investigate how the loss of transmission from CA1 pyramidal cells (PCs) impacts local circuit function, we employed CaMKIIα:Cre mice and virally expressed the tetanus toxin light chain (TeTX) in a Cre-dependent manner, specifically silencing synaptic transmission of CA1 PCs (Fig. 1a). Eight-week-old CaMKIIα:Cre mice, in which Cre expression is limited to CA1 PCs in the hippocampus[39,40], were bilaterally injected with Cre-dependent adeno-associated virus (AAV) co-expressing TeTX and a mCherry reporter (CA1-TeTX)[41], or the mCherry reporter alone (control) into the CA1 region. To assess the specificity of AAV-mediated transgene expression in PCs, mice were

sacrificed 15 days post-injection and immunohistochemistry was performed. mCherry expression was restricted to NeuN positive (neuron-specific marker) and GAD67 (interneuron marker) negative cells. Specifically, all 94 GAD67 positive cells, a marker for all interneuron types, analyzed were mCherry negative (44 & 50 cells from control & CA1-TeTX mice, respectively from 4 mice per condition) (Fig. 1b, c), whereas all 1328 mCherry positive cells (651 & 677 from control & CA1-TeTX mice, respectively) analyzed were NeuN positive, suggesting that TeTX expression is not present in inhibitory neurons, but restricted to PCs (Fig. 1b, c).

The efficacy of transmission blockade by TeTX, which cleaves synaptobrevin 2 and prevents neurotransmitter release[42], was verified in slices prepared from CA1-TeTX mice. Hippocampal coronal slices were prepared 12–15 days post-injection, and subiculum PCs, a postsynaptic target of CA1 PCs, were recorded to examine excitatory post-synaptic currents (EPSCs) originating from CA1-PCs input (Fig. 1d). In mCherry control slices, electrical stimulation of CA1 elicited robust EPSCs in subiculum PCs ($-115.72 \pm 31.33$, $-157.37 \pm 23.53$, $-161.65 \pm 32.83$, $-207.13 \pm 26.22$, $-287.12 \pm 77.02$, $-299.85 \pm 52.38$ pA (pico amp) for stimulation intensities of 1.10, 1.15, 1.20, 1.25, 1.30, 1.35 mA, respectively) (Fig. 1d, e). In contrast, evoked synaptic transmission was significantly reduced in CA1-TeTX slices ($-34.35 \pm 20.36$, $-57.69 \pm 30.67$, $-39.13 \pm 14.21$, $-72.90 \pm 24.55$, $-91.31 \pm 40.53$, $-71.71 \pm 21.30$ pA for stimulation intensity of 1.10, 1.15, 1.20, 1.25, 1.30, 1.35 mA, respectively) (Fig. 1d, e), demonstrating the high efficacy of TeTX in silencing synaptic transmission. Consistent with this, we also observed a reduction in the expression of synaptobrevin 2 in the subiculum in CA1-TeTX mice (Fig. 1f, g).

Next, to test whether the loss of transmission in CA1 PCs resulted in reduced inhibition, we prepared hippocampal coronal slices 12–15 days post-injection in CA1-TeTX and control mice, and CA1 PCs were recorded in whole-cell voltage clamp configuration with the holding potential set to +0 mV to examine spontaneous inhibitory post-synaptic currents (sIPSC) (Fig. 1h). We observed no difference in the cumulative fraction of sIPSC amplitude in neurons from CA1-TeTX and mCherry control mice (Fig. 1i). Similarly, we also did not find any significant difference in mean sIPSC amplitude (sIPSC amplitude/ neuron) between the groups (control, $35.47 \pm 3.56$ pA; CA1-TeTX, $41.78 \pm 2.44$ pA; $n = 12$–15 neurons, $N = 4$ mice/genotype) (Fig. 1h–j), suggesting that the response of CA1 neurons to incoming GABAergic input remains normal in CA1-TeTX mice. However, the frequency of sIPSC events in neurons from CA1-TeTX mice (404 sIPSC events) was dramatically lower than in mCherry controls (2490 sIPSC events) (Fig. 1k); thus, the mean sIPSC frequency (sIPSC frequency/neuron) was significantly lower in TeTX.mCherry ($0.07 \pm 0.01$ Hz) than control mice ($3.4 \pm 0.29$ Hz) (Fig. 1l). Together, these data suggest that the loss of synaptic transmission in CA1 PCs resulted in reduced inhibition without overtly affecting their response amplitudes. Given that in the CA1-TeTX mice the feedforward circuits are unperturbed, we reasoned that the reduction in spontaneous IPSCs in CA1 PCs is due to a loss of recruitment of feedback inhibition. To test this, we stimulated CA1 PCs and examined paired-pulse suppression (PPI), a measure of disynaptic inhibition commonly used to quantify FBI[43,44]. Specifically, a cohort of 8 weeks old CaMKIIα:Cre mice was bilaterally injected into the dorsal CA1 region with two Cre-dependent AAVs, one expressing channelrhodopsin 2 (ChR2) and the second expressing TeTX.mCherry or a mCherry control alone (Fig. 1m). Two weeks after the injection, hippocampal coronal slices were prepared and CA1 PCs were activated with blue light, while population spikes were recorded in CA1 (Fig. 1n, o). Optogenetic activation of CA1 PCs robustly induced a population response in CA1 in CA1-TeTX mice (Fig. 1o, p), demonstrating that the neurons remain electrically active. We found that the population spike amplitude ratios (paired-pulse ratio) were smaller in slices from control mice, indicating an overall suppression of population spikes during the second stimulation due to feedback inhibition. In contrast, in

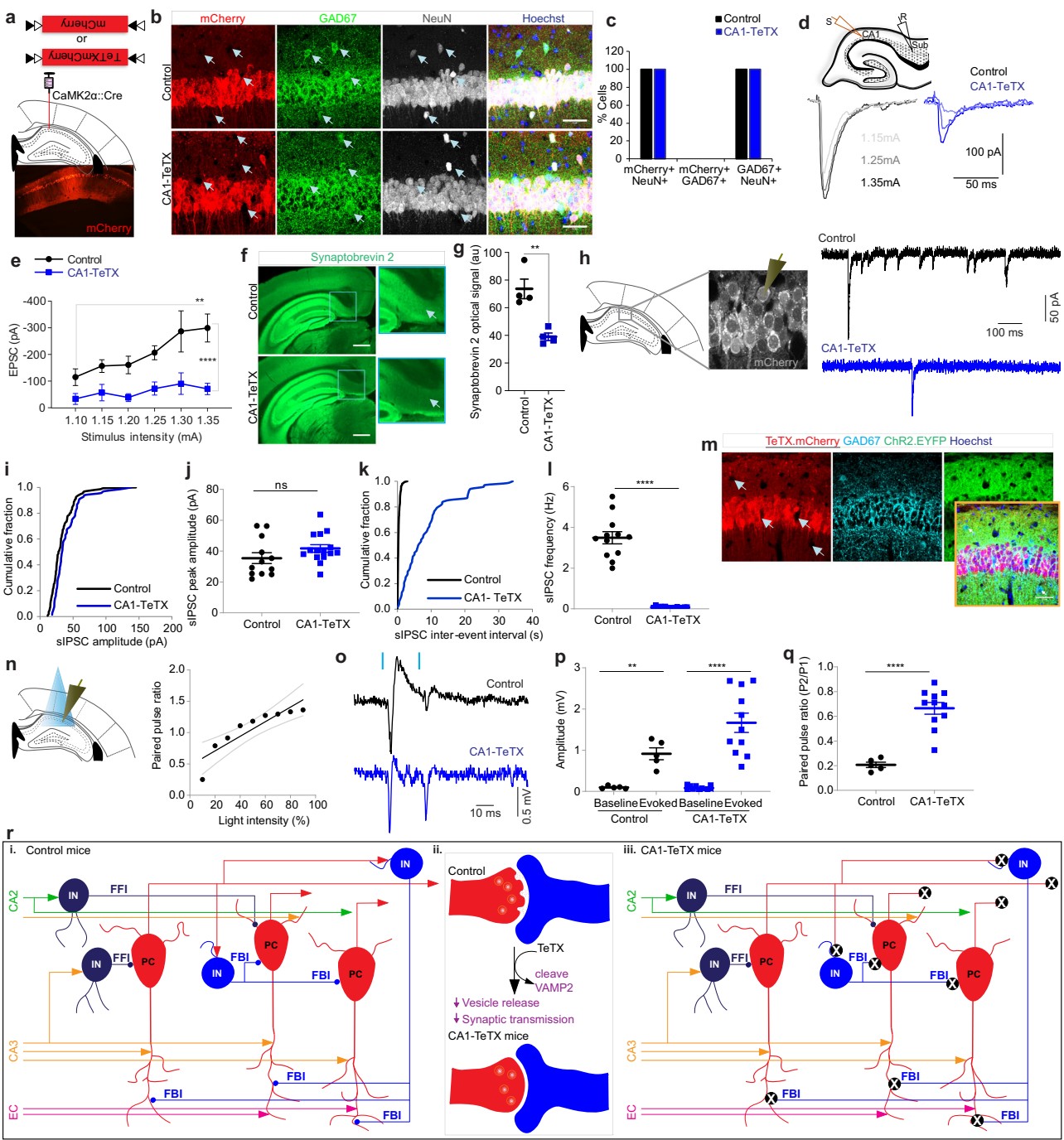

slices from CA1-TeTX mice, we observed a higher paired-pulse ratio (0.6651 ± 0.04) compared to controls (0.206 ± 0.02) (Fig. 1q and Supplementary Fig. 1a, b), suggesting that a loss of transmission in CA1 PCs resulted in reduced FBI (Fig. 1r). To further verify these findings, we electrically simulated CA1 with paired stimuli while recording population responses in CA1, again observing that CA1-TeTX mice exhibited an increased paired-pulse ratio compared to controls (Supplementary Fig. 1c, d). These results show that CA1 neurons retain normal electrical properties but have reduced FBI in CA1-TeTX mice (Fig. 1r).

## Loss of neurotransmission in CA1 pyramidal neurons enhances ripple power and frequency

The expression of tetanus toxin light chain in CA1 results in PCs that remain electrically active, and thus can generate spikes in response to natural depolarizing input but cannot release glutamate and recruit

FBI. This is an ideal model system to study the mechanistic role of FBI in high-frequency oscillations. To this end, CaMKIIα:Cre mice were bilaterally injected in dorsal CA1 with Cre-dependent AAV expressing TeTX.mCherry or mCherry alone as a control (Fig. 2a) and a microdrive containing eight adjustable nichrome-gold plated tetrodes was implanted targeting CA1 (Fig. 2a, b). Extracellular local field potentials (LFP) and population spiking activity were recorded as the mice sat quietly in a familiar chamber (rest/quiescence) or explored a linear track (Fig. 2c and Supplementary Fig. 2a–c).

We first examined the LFP spectral power profile between control and CA1-TeTX mice during rest, observing a dramatic increase in LFP power in the 100–400 Hz high-frequency range ($P < 0.0001$) in the CA1-TeTX mice (Fig. 2c, d). The increased 100–400 Hz power in CA1-TeTX mice was evident only during putative NREM (non-random eye movement) sleep, but not during REM epochs (Supplementary

**Fig. 1 | Loss of neurotransmission leads to reduced feedback inhibition in CA1 pyramidal cells. a** Experiment outline. Cre-dependent AAV expressing mCherry (control) or TeTX.mCherry (CA1-TeTX) was bilaterally injected into CA1 region of CaMKIIα::Cre mice. **b** Representative images show expression of mCherry reporter, inhibitory neuronal marker GAD67, general neuronal marker NeuN and Hoechst in CA1. Arrows indicate the absence of expression of reporter mCherry in GAD67::NeuN++ cells. Scale bar = 50 μm. **c** Plot shows the percentage of cells co-expressing indicated markers. All cells expressing mCherry are also positive for the general neuronal marker NeuN but are not positive for GAD67 ($N = 4$ mice/group). **d** Configuration of whole cell voltage clamp recording in acute hippocampal slices prepared from CaMKIIα::Cre mice previously injected with mCherry or TeTX.mCherry virus was assessed for changes in electrical stimulus-evoked CA1 to subiculum transmission. Example evoked EPSC traces are shown. Scale bar = 100 pA, 50 ms. **e** Electrical stimulation evoked EPSC amplitudes plotted as a function of stimulation intensity from slices with mCherry ($n = 9$ cells) or TeTX.mCherry ($n = 13$ cells, $N = 3$ mice/group) expressed in CA1 (two-way RM ANOVA between groups, $F(1, 119) = 52.18$, $P = 5.223 \times 10^{-11}$). **f** Example images show the expression of synaptobrevin 2 (inset shows cropped image). Arrows indicate the subiculum area. **g** A summary plot shows the optical signal of synaptobrevin 2 in the subiculum ($N = 4$ mice/group; two-sided $t$ test, $T = 4.582$, $P = 0.0038$). **h** Whole-cell voltage-clamp recording configuration in CA1 ex vivo brain slices. Representative spontaneous IPSC (sIPSC) traces are shown. **i** The cumulative fraction of sIPSC amplitude of CA1 pyramidal cells in control ($N = 4$ mice, $n = 12$ cells) and CA1-TeTX ($N = 4$ mice, $n = 15$ cells). **j** Mean sIPSC amplitude of CA1 pyramidal cells ($N = 4$ mice/group, $n = 12–15$ cells/group; unpaired two-sided $t$ test, $T = 1.50$, $P = 0.145$). **k** The cumulative fraction

of sIPSC inter-event interval frequency. **l** Mean frequency of sIPSC of CA1 pyramidal cells ($N = 4$ mice/group, $n = 12–15$ cells/group; unpaired two-sided $t$ test, $T = 12.83$, $P = 1.69 \times 10^{-12}$). **m** Representative images show expression of mCherry reporter, GAD67, ChR2 tagged with EYFP, and Hoechst in CA1. **n** Recording configuration in CA1 ex vivo brain slices. A stimulus-response curve with a trendline and confidence interval shows changes in the population spike amplitude in response to varying intensities of the optogenetic stimulus (Pearson correlation, $R^2 = 0.7902$, $P = 0.0013$). **o** Representative extracellular population spike responses from the CA1 pyramidal cell layer in response to optogenetics activation of CA1 pyramidal neurons from slices from mice co-expressing mCherry & ChR2 (control) or TeTX.mCherry & ChR2 (CA1-TeTX) ($n = 5–11$ slices, $N = 4$ mice/genotype). **p** Plot shows optogenetically evoked extracellular population spike amplitude ($N = 4$ mice/group, $n = 5–11$ slices/group, paired $t$ test; control, $T = 5.779$, $P = 0.0045$; CA1-TeTX, $T = 6.532$, $P = 6.625 \times 10^{-5}$). **q** Paired pulse ratio in control and CA1-TeTX ($N = 4$ mice/group, $n = 5–11$ slices/group, unpaired two-sided $t$ test, $T = 6.276$, $P = 2.035 \times 10^{-5}$). **r i** Schematic of a simplified CA1 neural circuitry. Inhibitory neurons (INs) in CA1 that receive input from upstream brain areas directly regulate the CA1 excitatory neurons (PCs) mediate feedforward inhibition (FFI), whereas inhibitory neurons in CA1 that receive input from CA1 PCs feedback onto CA1 PCs and regulate the activity of PCs in CA1 mediate feedback inhibition (FBI). **ii** Expression of TeTX in CA1 PCs impairs synaptic transmission in CA1-TeTX mice. **iii** Schematic shows the reduction in FBI in CA1-TeTX mice. Data shown in (**e, g, j, l, p, q**) represent mean ± standard error of the mean (s.e.m.). ** and **** indicate $P < 0.01$ and $P < 0.0001$, respectively. ns not significant, au arbitrary units. Source data are provided as a Source Data file.

Fig. 2d). Therefore, we next systematically analyzed high-frequency oscillations, specifically examining ripples, which occur predominantly during slow wave NREM sleep. We applied a ripple detection algorithm, detecting events in the 80–400 Hz frequency range whose amplitudes exceed 4 times the standard deviation of mean power and lasted for more than 30 ms (Fig. 2e). We found that the ripple amplitude in CA1-TeTX mice ($1.51 \pm 0.20$ mA) was drastically higher than in control mice ($0.306 \pm 0.03$ mA) (Fig. 2f), suggesting that the loss of PC output increases the power of ripple oscillations in CA1. In addition, we found that mean ripple frequency ($178.2 \pm 4.19$ Hz; $n = 19838$ ripples) (Fig. 2g) and duration ($114.7 \pm 4.56$ ms) (Fig. 2h) in CA1-TeTX mice were also significantly larger when compared to controls (freq: $138.8 \pm 3.58$ Hz; duration: $100 \pm 2.95$ ms; $n = 13504$ ripples); however no change in the interval between ripple events was observed ($2.04 \pm 0.24$ and $2.614 \pm 0.49$ events/s in CA1-TeTX & control mice, respectively) (Fig. 2i). Concurrent with the increase in ripple amplitude and frequency, on the level of population activity, we observed a significant increase in ripple-related population spiking in CA1-TeTX mice ($170.0 \pm 14.89\%$) compared to controls ($100 \pm 11.24\%$) (Fig. 2j).

### Increased theta power and theta phase-locked high-frequency activity during active exploration in CA1-TeTX mice

How does the loss of transmission in CA1 PCs affect oscillations during movement? On the linear track (i.e., active exploration), control ($60.644 \pm 10.29$ m) and CA1-TeTX mice ($63.924 \pm 13.71$ m) demonstrated similar behavior, covering comparable distances on average across the sessions with no obvious behavioral abnormalities (Supplementary Fig. 3a). Power in the theta ($7–12$ Hz) band was significantly elevated in the CA1-TeTX mice compared to controls, but both low ($30–50$ Hz) and high ($50–100$ Hz) gamma power was comparable across the groups (Fig. 3b). The elevated theta peak-to-peak amplitude in the CA1-TeTX was independent of changes in waveform characteristics and cycle duration, which were similar to controls (Fig. 3c, d), suggesting that incoming input to CA1 is sufficient to drive theta oscillations. As we observed during rest, during movement the mean population spike rate significantly increased in CA1-TeTX mice ($132.3 \pm 7.982\%$) compared to control mice ($100.0 \pm 9.39\%$; $T = 2.636$, $P = 0.010$) (Supplementary Fig. 3b); however, the autocorrelation of the spike trains revealed a lag of ~125 ms between the peaks in both the control and CA1-TeTX mice

(Supplementary Fig. 3c), suggesting that CA1 populations in both groups spike rhythmically at theta frequency.

These observations prompted us to next examine whether theta-gamma modulation was impacted in CA1-TeTX mice. We first examined the phase-amplitude coupling of low and high gamma to theta and observed no differences between groups (Supplementary Fig. 3d, e). We then asked if the previously reported[45] positive correlation between mice movement velocity and LFP gamma power remained intact in CA1 and observed a strong positive correlation in both the control (adjusted $R^2$, $0.808 \pm 0.045$) and CA1-TeTX mice (adjusted $R^2$, $0.857 \pm 0.041$) (Supplementary Fig. 3f–j). Moreover, we also observed a positive relationship between the power of high-frequency activity ($100–400$ Hz) and the animal movement velocity in both controls (adjusted $R^2$, $0.815 \pm 0.045$) and CA1-TeTX mice (adjusted $R^2$, $0.776 \pm 0.054$) (Supplementary Fig. 3f–j), suggesting that the CA1 PC output may not play a crucial role in driving gamma with respect to theta phase in CA1 and the feedforward circuit is likely sufficient to drive these oscillatory activities in the CA1.

### Theta phase locked spatially restricted population spikes in CA1-TeTX mice

While power in the lower frequency range (~20–100 Hz) during movement were largely similar between the two groups of mice, visual inspection of the LFP waveforms suggested that the increased high-frequency power obvious in the PSD reflected the presence of epileptic-like pathophysiological population discharge events in CA1-TeTX mice (Fig. 3b, e). To further investigate these events, we detected them in a manner similar to that used to detect ripples, increased power in the 80–400 Hz frequency band. Indeed, we observed high-frequency oscillatory events (HFEs), present only in CA1-TeTX mice, occurring at an average frequency of $23.24 \pm 7.62$ events/min (Fig. 3e and Supplementary Fig. 4a). These HFEs ($177.76 \pm 0.417$ Hz, $n = 8466$ HFEs) were present during ongoing theta (Fig. 3e, f) and both their occurrence and amplitude (theta-ripple modulation index; $0.0079 \pm 0.0016$) were significantly modulated by theta phase (Fig. 3g), with the HFEs tightly locked to the trough of the slow oscillation (Fig. 3e–h).

High-frequency oscillations, such as ripples across the CA1 region, are typically anisotropic, with the co-occurrence of ripples decreasing as the distance in electrode locations increases within CA1[46]. We

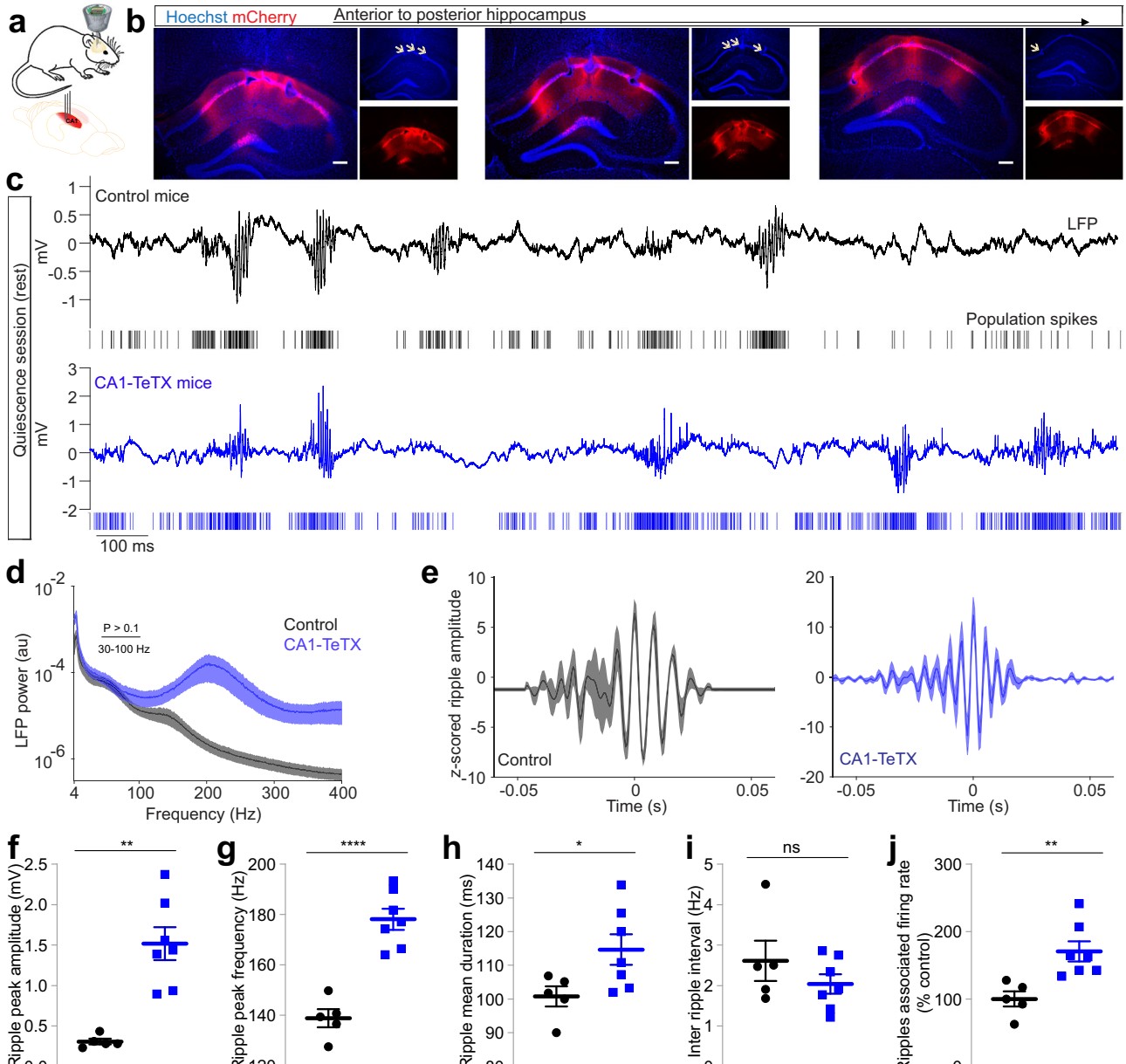

**Fig. 2 | Enhanced amplitude and frequency of ripples in CA1-TeTX mice.**
**a** Recording configuration in CA1 in vivo. Cre-dependent AAV expressing mCherry or TeTX.mCherry was bilaterally injected into CA1 region of CaMKIIα::Cre mice, and mice were also implanted with microdrive. After recuperation from surgery and viral expression, mice were habituated in a recording box and electrophysiological recording was performed. **b** Example images show post-recording lesioned electrode tip sites (arrow) and mCherry reporter expression in CA1 from CA1-TeTX mice. **c** Representative LFP traces and population spikes from control and CA1-TeTX mice. **d** LFP power profile (mean ± s.e.m.) in control and CA1-TeTX mice ($N = 5$ control and 7 CA1-TeTX mice; two-way RM ANOVA, frequency x genotype interaction, $F_{(802,8020)} = 6.491$, $P < 0.0001$). **e** Z-scored ripple waveforms from control (left) and CA1-TeTX (right) mice. **f–i** Mean amplitude (**f**; $N = 5$ control and 7 CA1-TeTX mice; unpaired two-sided $t$ test, $T = 4.927$, P = 0.0006), frequency (**g**; $T = 6.727$, $P = 5.184 \times 10^{-5}$), duration (**h**; $T = 2.318$, $P = 0.042$), and incidence interval (**i**; $T = 1.137$, $P = 0.281$) of ripples. **j** Mean ripples associated population spike rate ($T = 3.506$, $P = 0.0057$). Data shown in (**e–j**) represent mean ± s.e.m. *, **, and **** indicate $P < 0.05$, $P < 0.01$, and $P < 0.0001$, respectively. ns= not significant. Source data are provided as a Source Data file.

therefore asked whether HFEs were synchronized across CA1 during active exploration in CA1-TeTX mice. The frequency and theta phase of HFEs were examined from simultaneous recordings at five locations in CA1, spaced at least 300 μm apart (Fig. 4a–c). We observed that the co-occurrence of HFEs between CA1 areas was rare, despite the fact that HFE frequency and preferred theta-phase was consistent as all locations (Fig. 4a–c). To quantify this and compare it to ripple events in both groups, we formulated a cumulative co-occurrence index (CCI; see "Methods"; Supplementary Data 1) whereby we tested the probability of HFEs occurring simultaneously in at least two locations (different tetrodes) in the CA1-TeTX mice. Interestingly, the CCI of ripples

was comparable between control (0.1439 ± 0.034) and CA1-TeTX mice (0.1568 ± 0.023) during the rest session. However, we found that in the CA1-TeTX mice, the CCI of HFEs during active exploration (HFEs; 0.08696 ± 0.009) was significantly reduced compared to ripples during the quiescence session (Fig. 4c). We next reasoned that the amplitude correlations across tetrodes would also be lower during exploration than rest. Thus, we examined the relationship between the mean peak amplitudes of HFEs detected in any given electrode and the corresponding peak amplitudes of all other channels. The mean peak phase and amplitude correlations across CA1 locations during HFEs significantly differed between active exploration and the quiescence

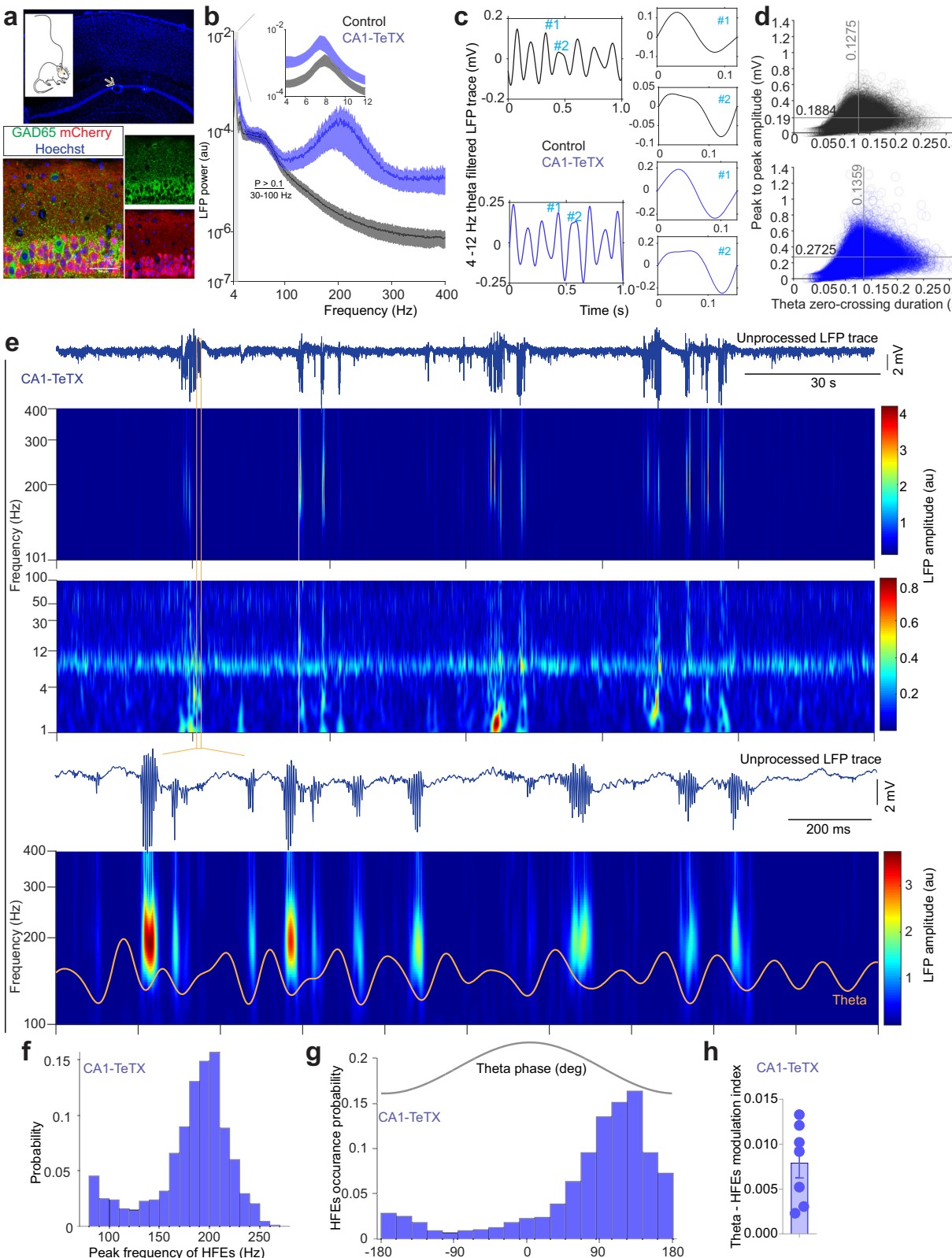

session in CA1-TeTX mice (Fig. 4d–f Supplementary and Fig. 5a–c), suggesting a reduced relationship of high-frequency events between CA1 locations (tetrodes) during movement.

As the HFEs were sporadic and local events, we next separated theta cycles into two classes based on the presence or absence of HFEs. The mean spike rate in theta cycles with HFEs was significantly higher than in theta cycles without HFEs across all tetrodes in CA1-TeTX mice

(Fig. 4g). How does the increase in the spike rate affect LFP theta and spiking rhythmicity? To understand this, we performed autocorrelation of spiking. We observed a lag of ~125 ms between the peaks in the autocorrelogram in theta cycles with no HFE, closely following the mean LFP theta frequency (Fig. 4h). Interestingly, spikes also showed a clear peak at ~125 ms lag in the autocorrelogram generated from theta cycles with HFEs, albeit with a bit stronger spiking probability than

**Fig. 3 | Theta phase-locked high-frequency activity during active exploration in CA1-TeTX mice. a** Confocal images show the recording site (top, arrow; Hoechst in blue), reporter mCherry (red) expression, and interneuron marker GAD67 (green) in CA1 from control mice. **b** LFP theta power profile (mean ± s.e.m.) during active exploration (two-way RM ANOVA). **c** Example theta-filtered traces and waveforms extracted based on zero-crossings are shown. **d** Plots show the duration and the peak-to-peak amplitude of theta cycles in control (0.1884 mV) and CA1-TeTX mice (0.2725 mV). **e** Representative raw LFP trace and the corresponding amplitude scalogram from CA1-TeTX mice during linear track exploration. Bottom, 4–12 Hz theta filtered trace is superimposed on the LFP scalogram. **f** Histogram shows the peak frequency of high-frequency events (HFEs) detected in CA1-TeTX mice during active exploration. **g** Histogram shows the phase of theta at which HFEs occur in CA1-TeTX mice ($N = 7$ mice). The relationship between theta phase and HFEs occurrence was statistically significant (Rayleigh z test, $P = 0.0021$). **h** Plot shows the modulation index of theta phase and amplitude of HFEs (one-sample two-tailed $t$ test, $T = 4.760$, $P = 0.0031$). Error bar shows s.e.m. Source data are provided as a Source Data file.

during the theta cycles with no HFEs (Fig. 4h). Moreover, despite the increase in population spiking and the appearance of HFEs, we observed no difference in the theta frequency between theta cycles with and without HFEs (Fig. 4i).

Previous studies have shown that the spiking of CA1 pyramidal cells is most prominent at the trough of theta cycles[47]. This phase preference is preserved in the CA1-TeTX mice, indicating that FBI may play a negligible role in modulating the relationship between the theta phase and population spiking in CA1. On the other hand, these results suggest these population spike bursts correlate with HFEs during navigation and that the sparsity of ensemble activity is disrupted in CA1-TeTX mice. Consistent with CCI of HFEs during linear track, time-resolved frequency assessment of LFP by continuous wavelet transforms revealed uncorrelated bursts of HFEs events between simultaneously recorded tetrodes in CA1 in CA1-TeTX mice during linear track exploration (Fig. 4j), suggesting local nature of HFEs within CA1. Therefore, we explored the spatial properties of population spikes on a tetrode-by-tetrode basis. We constructed place field maps using the timestamps of the peak magnitude of HFEs associated population spike bursts and observed that HFEs recorded on different CA1 channels occur preferentially at unique locations on the linear track and the spatial locations of the events remained stable within a session, resembling the activity typically seen in single place cells in CA1 (Fig. 4k and Supplementary Figs. 5d and 6a, b). These data are consistent with the importance of lateral inhibition in shaping local sparsity in the CA1 spatial representation[36] and provide a clear evidence of the consequence of a loss of sparsity on a population level.

## Discussion

The unique contributions of the local feedforward and feedback inhibitory circuits to CA1 physiology and function has been difficult to untangle due to the fact that many interneurons receive excitatory drive from both external and internal inputs, as well as due to the heterogeneity in function among genetically defined subclasses of interneurons[33,48–50]. Here we leverage our ability to chronically inhibit neurotransmitter release specifically from CA1 pyramidal cells while maintaining their ability to generate action potentials in response to input, a necessity in assessing the impact of manipulation of population activity. Given the local circuit architecture, the primary result of this manipulation is to selectively remove the ability of these neurons to recruit feedback inhibition to their neighbors (Fig. 1l, q). This manipulation led to the appearance of unique forms of hyperexcitability-linked pathophysiology, both during rest and during movement. Sharp-wave ripples (SWRs), thought to be triggered by input from CA3 and/or CA2, and sculpted by local excitatory-inhibitory circuits[6,51], became larger in amplitude, longer in duration and faster in frequency (Fig. 2c–j). All these changes are consistent with an increase in the number of CA1 pyramidal cells participating in their generation and a role for FBI in terminating the events[30,52,53]. Similarly, high-frequency oscillations appeared during locomotion, tightly locked to the trough of theta oscillation (Fig. 3e–h), and surprisingly, demonstrating properties consistent with those seen in single place fields (Fig. 4k).

Ripples are among the largest and fastest oscillations naturally occurring in the mammalian brain[6]. They have been shown to play key roles in temporally organizing the underlying spiking of CA1 pyramidal neurons and play roles in memory consolidation, decision making and planning[14–16,18]. One recent key observation is that although ripples precisely entrain the spike timing of pyramidal cells[54], the fraction of CA1 neurons spiking during a given ripple is very sparse, roughly 10%[55], with a log-normal distribution of participation, such that a small fraction of PCs participate in over 50% of the ripple events. Given that we observed no difference in the frequency of occurrence of ripples, these data are consistent with the current understanding that the feedforward excitation and inhibition resulting from CA3/2 input is sufficient to trigger these events and FFI and intrinsic CA1 PC properties can produce a fast oscillation. The crucial differences in the absence of FBI we observed were in the amplitude, frequency, and duration of ripples, which all showed significant increases. This suggests that FBI plays a key role in controlling the sparsity of the CA1 pyramidal cells recruited, and in its absence, the typical excitatory drive arriving from CA3/2 results in oscillations that resemble pathophysiology typically seen in an epileptic brain[56,57].

During locomotion we observed no changes in the frequency of the theta oscillation; however, there was a significant increase in the amplitude, inconsistent with some predictions from previous modelling[58]. The underlying mechanism leading to this amplitude increase may reflect the increase in pyramidal cell spiking in CA1 resulting from a loss of lateral/feedback inhibition (Fig. 1q and Supplementary Fig. 3b), as previous work has shown that the elimination of a fraction of CA1 pyramidal cells reduces theta amplitude[59,60]. Additionally, we must consider the inability of CA1 pyramidal cells to recruit local inhibitory neurons which target the medial septum (MS)[61–63]. While the MS targeting GABAergic neurons are thought to play a role in modulating theta frequency via reciprocal connections with inhibitory MS neurons[64–66], we cannot eliminate the possibility that this may also influence cholinergic drive and, as a result, the power of theta. In contrast, we saw no change in the frequency, amplitude or phase-amplitude coupling of both slow and fast gamma oscillations during rest and movement. This observation is consistent with the hypothesis that these faster oscillations, primarily present during movement, are driven by coupling with oscillations driven by CA3 (slow gamma) and entorhinal cortex (EC; fast gamma)[26,27]. These observations also suggest that the CA3/2 and EC inputs to CA1 were unlikely altered in CA1-TeTX mice. Furthermore, given the clear evidence of NREM-specific ripples, and REM- and movement-specific theta, it is unlikely that CA1-TeTX mice have excitatory /inhibitory (E/I) imbalance across the whole hippocampus. A recent study demonstrated that there is a risk that the CaMKIIα promoter, particularly short versions designed to directed expression when packaged in AAV, can drive expression of AAV payloads in interneurons in cortical networks, sometimes at levels not detectable via co-expression of a fluorophore[67]. However, here we employ a well-characterized transgenic mouse line, in which a much larger CaMKIIα promoter fragment directs Cre expression specifically to excitatory neurons[40]. Further, in our experimental design, Cre-dependent expression of TeTX and/or mCherry is driven by the robust EF1a promoter, thus the observation of no co-localization of mCherry expression with known interneuron markers (Figs. 1b, c, m and 3a) following viral injection, as well as the specific nature of oscillatory defects observed suggests TeTX expression is negligible in

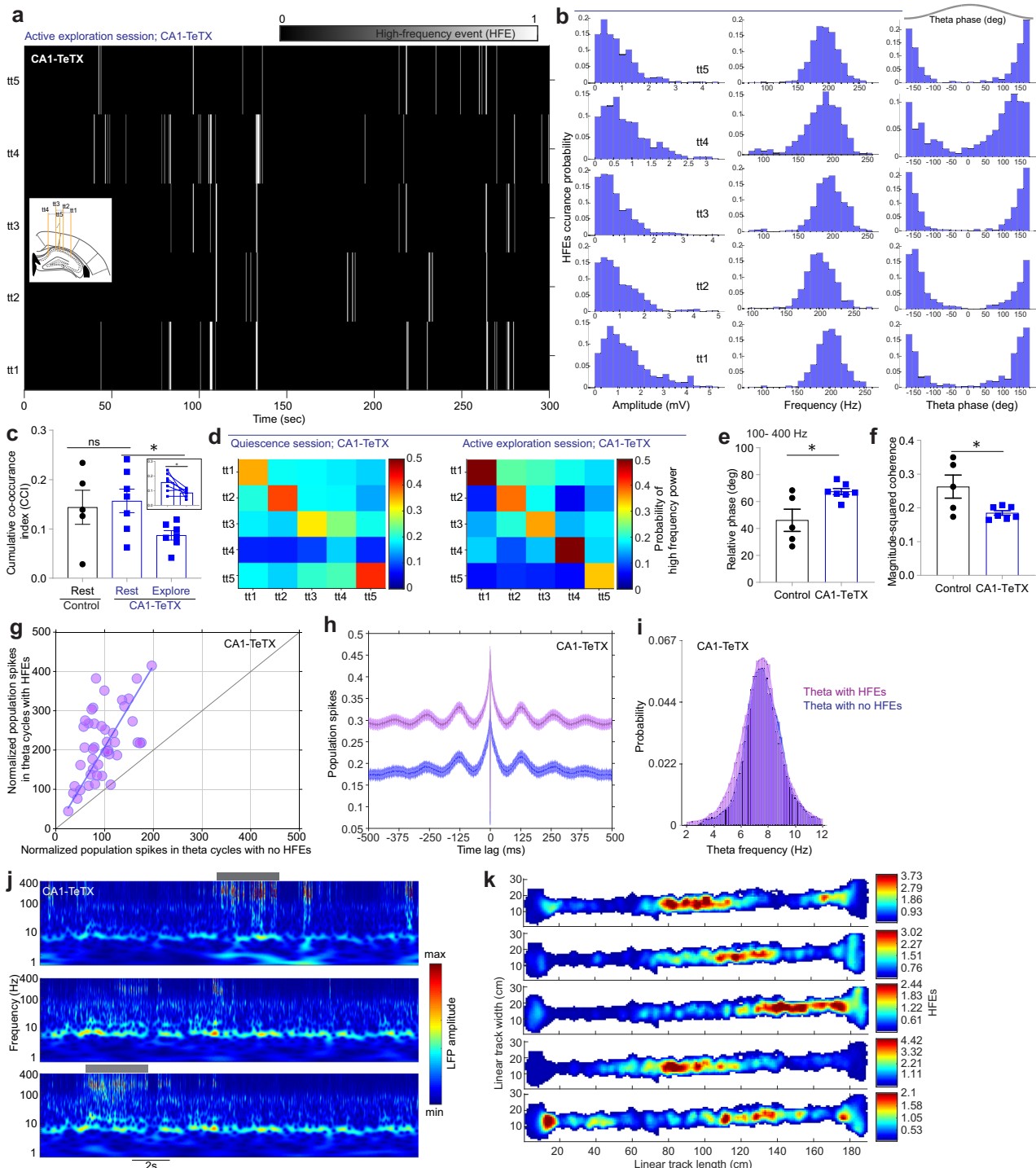

**Fig. 4 | Local nature of high-frequency events in CA1-TeTX mice. a** Heatmap shows HFEs in the time domain during active exploration sessions in all five tetrodes in CA1-TeTX mice. Inset cartoon shows tetrodes (tt) configuration and placement in CA1. **b** Plots show the peak amplitudes & frequency of HFEs and the theta-phase at which HFEs occur in all five tetrodes (corresponding tetrodes as in (**a**)) in CA1-TeTX mice during active exploration sessions. **c** Cumulative co-occurrence index of HFEs during quiescence and active exploration sessions in control and CA1-TeTX mice ($N = 5$ control and 7 CA1-TeTX mice; unpaired two-sided $t$ test, $T = 0.3184$, $P = 0.756$). Inset plot shows pairwise comparison in CA1-TeTX mice (paired $t$ test, $T = 2.563$, $P = 0.0428$). **d** Plots show the probability of LFP power of high-frequency during rest (left; ripples) and active exploration sessions (right; HFEs) in CA1-TeTX mice. **e, f** Plots show the relative phase difference (**e**; unpaired two-sided $t$ test, $T = 2.881$, $P = 0.0164$) and the magnitude-squared coherence

(**f**; unpaired two-sided $t$ test, $T = 2.632$, $P = 0.0251$) of 100−400 Hz across all tts between control and CA1-TeTX mice ($N = 5$ control and 7 CA1-TeTX mice). **g** Plot shows the mean population spike rate during theta cycles with or without HFEs in CA1-TeTX mice. **h** Gaussian-smoothed autocorrelogram of population spiking (mean ± s.e.m.) in theta cycles with (purple) and without HFEs (blue) in CA1-TeTX mice. **i** Mean theta frequency in cycles with or without HFEs in CA1-TeTX mice. **j** A 20 s window from a CA1-TeTX mouse showing three simultaneously recorded LFP scalograms during linear track exploration. HFEs bursts are shown to the top (grey line). **k** Place map shows a rate of HFEs bursts time plotted by positional bins of the linear track. Each row corresponds to one CA1 location (tt) from a representative CA1-TeTX mouse. Data shown in (**c, e, f**) represent mean ± s.e.m. * indicates $P < 0.05$, ns not significant. Source data are provided as a Source Data file.

interneurons and FFI remains intact. Nonetheless, it will be interesting in future work to examine the impacts of acute loss of FBI on hippocampal oscillations.

One unfortunate consequence of the size and frequency of the pathophysiological events in the CA1-TeTX mice, both during rest and movement, was the saturation of conventional amplification, precluding the recording of single unit activity. Thus, while we could examine multiunit spiking, it was impossible to adjust the amplification to a level that allowed the clustering of single neurons. This limits our ability to further investigate why the HFEs occurring during theta appear highly similar to single place cells, spatial in nature and stable across laps (Fig. 4k and Supplementary Figs. 5d and 6a, b), as well as to precisely estimate the change in the participation of single units during ripples. During locomotion, the spatial nature of the local HFEs may reflect the track location at which CA3 input to that part of CA1 is at its peak, consistent with the strong locking of the events to the trough of theta. While in a normal brain, this input would trigger the formation of one or several strong place cell(s) at that location[68], in the absence of lateral inhibition, a larger population of neurons generate spikes in response, leading to local hyperexcitability and the generation of a fast oscillation. One consequence of place cell activity is the recruitment of lateral feedback inhibition to other pyramidal cells in the surrounding area[36]. This ensures sparsity of the representations and suggests a modular organization of the spatial representation across CA1, defined by the extent of the lateral inhibitory networks. While we cannot estimate the true extent of these local networks from our data, given that the spatially triggered HFEs were distinct, even on adjacent tetrodes spaced roughly 300 μm apart, this puts an upper bound on their size. Further the spatial nature of the HFEs we observed in the CA1-TeTX mice is reminiscent of similar events we previously observed in the CA1 of mice with a chronic blockade of CA2 excitatory transmission (CA2-TeTX mice)[69]. An important distinction, however, is that in the CA2-TeTX mice the spatially triggered hyperexcitability appeared simultaneously across all recording sites, consistent with the hypothesis that those events were driven by increased CA3 input resulting from a loss of CA2 recruited FFI in the recurrent CA3 network. In contrast, the HFEs we observed in the CA1-TeTX mice, while similar in their spatial extent and stability, were highly localized to small areas of CA1, reflecting the restricted influence of the lateral feedback inhibition in the CA1 circuit. Thus, while chronic synaptic silencing, such that mediated by the tetanus toxin light chain (TeTX), has its limitations, it has proven to be a valuable tool in dissecting the capabilities and consequences of specific changes in local circuit function[26,41,42,69–71].

## Methods

### Experimental animals
Young adult (8-weeks-old) transgenic mice of both sex (male & female) with Cre-recombinase expressed in excitatory neurons in hippocampal CA1 (CaMKIIα::Cre; JAX#005359) were used for all experiments. CaMKIIα::Cre mice were bred and maintained in our animal facility, and were genotyped. Number of mice used in each experiment is listed in main text and/or figure legends accordingly, and experiments were done using age-matched wildtype littermates and sex balanced. Mice were housed in groups of two to five under a 12 hr light–dark cycle (light cycle, 7 AM–7 PM). Animals were given ad libitum access to water and laboratory mouse diet at all times. After microdrive implantation surgical procedures, the mice were single housed. All experiments were approved by RIKEN Animal Care and Use Committee, or the Committee for Animal Care of the Division of Comparative Medicine (DVM) at the Massachusetts Institute of Technology (MIT), conformed to NIH and institutional guidelines.

### Viral Injections
**AAV injections for input-output curve, sIPSC and in vivo neural recording experiments.** AAV-EF1a-DIO-mCherry or AAV-EF1a-DIO-

Tetanus toxin-mCherry (see refs. 41,69 for AAV vector construction) was injected into CaMKIIα::Cre mice. briefly, 8-week old mice were anesthetized with avertin (2, 2, 2-tribromoethanol; Sigma-Aldrich, # T4840 476 mg/kg b.w, i.p) and restrained in a stereotactic apparatus (Steolting). We made an incision above the skull, and holes were drilled on the skull bilaterally in the following coordinates with reference to bregma: A/P = −2.0 mm, L/M = ±1.45 mm. Cre-dependent AAV expressing either mCherry control (AAV-DIO-mCherry; $6 \times 10^8$ viral particles/ml) or tetanus toxin light chain fused with an mCherry reporter (TeTX; AAV.DIO-Tetanus toxin-mCherry; $6 \times 10^8$ viral particles/ml) was infused (50 ηl per min) with 500 ηl per hemisphere bilaterally into the CA1 under the control of a syringe pump (Micro UMP, World Precision Instruments). The virus was infused using a 10 μl NanoFil syringe fitted with a 35 gauge needle (World Precision Instruments) lowered down to the CA1 (D/V = −1.4 mm). Following injections, we waited for an additional 10 min before slowly withdrawing the needles over 5 min. The incisions were closed by suturing and mice were allowed to recover.

**AAV injections for paired-pulse-ratio experiment.** We performed stereotactic procedure as described above and bilaterally injected 1000 ηl of a mixture (500 ηl each) of Cre-dependent AAV expressing ChR2-EYFP (AAV-EF1a-DIO-hChR2(H134R)-EYFP, $3.2 \times 10^{12}$ virus molecules/ml (UNC vector core)) and AAV-DIO-mCherry or AAV-DIO-Tetanus toxin-mCherry into dorsal CA1 in CaMKIIα::Cre mice (8-weeks-old).

### Immunohistochemistry
**Tissue preparation.** After the experiments, the animals were deeply anesthetized with avertin and transcardially perfused with 0.1 M phosphate-buffered saline (PBS, pH 7.4), followed by 4% paraformaldehyde (PFA, Electron Microscopic Sciences) in PBS. The brains were removed and postfixed overnight at 4 °C with 4% PFA, before they were rinsed in PBS for 5 min and kept at cold room until further analysis.

Coronal sections of 40 μm thickness were prepared using a vibratome (Leica), collected in 24-well plates and stored at 4 °C until further processing. For examining endogenous reporter expression and verifying lesioned site of recording locations, slices were washed in PBS and then mounted with VECTASHIELD Mounting Medium with DAPI (Vector Laboratories, Cat#: H-1500; RRID AB_2336788). For GAD67, NeuN and synaptobrevin 2 (VAMP2) staining, sections were first blocked with blocking solution (PBS + 0.05% triton X-100 + 5% normal donkey serum), incubated overnight in a primary antibody prepared in blocking solution (mouse α-GAD67 (1:1000 dilution), Sigma-Aldrich, Cat# MAB5406; guinea pig α-Neun (1:1000), Synaptic Systems, Cat# 266 004; rabbit α-VAMP2 (1:250), Synaptic Systems, Cat# 104 202), followed by three washes (10 min each) with PBS + 0.05% triton X-100 and subsequently stained with a suitable Alexa Fluor- conjugated secondary antibody prepared in blocking solution (1:1000; Thermo Fisher Scientific, Cat# A21202, Cat# A-21450, Cat# A-31573) for 2 h at room temperature. Following three washes (15 min each) with PBS + 0.05% triton X-100 containing Hoechst, slices were mounted with Fluromount G, and coverslipped. mCherry and EYFP reporter signals were imaged without further amplifying with antibodies. Images were acquired using either LSM 710 (Zeiss) confocal microscope or DM6000B (Leica) epifluorescent microscope with 5×, 10×, or 40× objective at identical settings across experimental groups. Representative images shown were prepared using Image J.

### Ex vivo electrophysiology
Acute hippocampal slice recordings were conducted from control and/or CA1-TeTX mice after appropriate viral injections and recovery. Mice were anesthetized with avertin and decapitated. Coronal hippocampal slices (400 μm thick) were prepared in ice-cold cutting solution (in mM: 211 sucrose, 3.3 KCl, 1.3 NaH$_2$PO$_4$, 0.5 CaCl$_2$, 10 MgCl$_2$, 26 NaHCO$_3$ and 11 glucose) using a Leica VT1000S vibratome (Leica).

Slices were recovered with artificial cerebrospinal fluid (ACSF) consisting of (in mM) 124 NaCl, 3.3 KCl, 1.3 $NaH_2PO_4$, 2.5 $CaCl_2$, 1.5 $MgCl_2$, 26 $NaHCO_3$ and 11 glucose for 1 h at 32 °C, and subsequently were held at room temperature. The cutting and recording solution were saturated with 95% $O_2$/5% $CO_2$ and maintained at a pH of 7.4 for 1–4 h. Individual slices for recording were then transferred to a submerged recording chamber and perfused with ACSF at a constant rate of 2–2.5 ml/min at room temperature. A borosilicate glass electrode (resistance of 6-7 MΩ) with pipette solution containing (in mM) 145 CsCl; 5 NaCl; 10 HEPES; 10 EGTA, 4 MgATP, 0.3 $Na_2GTP$ was used. Recordings were obtained using MultiClamp 700B (Molecular Devices). Data were low-pass filtered at 2 kHz, digitized using DigiData 1440 A, and acquired using pClamp10 at 10 kHz sampling frequency.

**Evoked EPSC.** Electrical stimulation of CA1 afferents was applied at 0.05 Hz in the stratum radiatum with increment of 0.05 mA current. Subiculum pyramidal neurons were visualized under the miscroscope and whole cell voltage clamp recordings of subiculum pyramidal cells were performed. EPSC were recorded at a holding potential of −70 mV.

**Spontaneous IPSC.** Hippocampal CA1 pyramidal neurons were patched and whole cell voltage clamp recordings were performed. sIPSC were recorded for a minimum of three minutes at a holding potential of +0 mV in the presence of 10 μM CNQX and 50 μM AP-5. Recordings were discontinued if series resistance increased by more than 20%.

**Paired-pulse ratio.** CA1 population field potentials evoked by optogenetic stimulation (473 nm blue light controlled by a CoolLED model PE-100) driven by a TTL pulse. Recording electrode was placed in stratum pyramidale layer in CA1 and optimized for maximal evoked responses. Paired-pulse responses were evoked with a 20 ms interval between optogenetics pulses of 2 ms duration, averaged over 5 sweeps each (0.1 Hz, monophasic pulse). Our initial optimization showed that light intensity levels (20%) reliably produced population spike amplitude of ≥0.5 mV, we thus used 20% light intensity throughout the experiments for all recordings.

### In vivo electrophysiology

**Microdrive.** Microdrives were prepared in-house with 3D print base with the assistance of the Advanced Manufacturing Support Team, RIKEN Center for Advanced Photonics, Japan. Tetrodes were also prepared in-house using nichrome wires (California fine wire company) and arranged in two rows of 4 tetrodes, each of them running along the CA1 medial-to-lateral axis of the dorsal hippocampus. Tetrodes were gold (Gold plating non-cyanide solution, Neuralynx) plated to an impedance of 175 to 250 kΩ using stimulus isolator (A365 World Precision Instruments) and electrode impedance tester (IMP-2A, Bak Electronics Inc). Tetrode wires were individually connected to the electrode interface board (EIB-36, Neuralynx) affixed to the microdrives. Buffered 36 channel headstage with Omnetics 44 connector (HS-36-LED, Neuralynx) was connected to the EIB-36 and tethered to the recording system (Digital Lynx SX, Neuralynx).

**Surgery.** Mice were deeply anaesthetized with avertin and affixed in stereotaxic apparatus. AAV-injected mice were implanted with a microdrive array consisting of eight independently adjustable nichrome tetrodes (14 μm diameter) targeted to the CA1 in the right hemisphere. We first bilaterally infused AAV, and 20 min later mice were implanted with microdrive. Specifically, using a dental drill, a 2 × 4 mm piece of skull was removed, which was above the right hippocampal CA1 (with the center of the craniotomy around the stereotaxic coordinates relative to bregma; A/P = −2.0 mm, L/M = ± 1.45 mm). Immediately after the craniotomy, a drop of saline was applied on the dura to keep the dura intact and moist. We did not remove/modify dura and kept it intact throughout the procedure. Following the skull

removal from above CA1, we made two additional drilling holes above the frontal cortex and one above the parietal area in the left side and placed three skull screws. Tetrode drives were then fitted to the stereotactic apparatus and aligned to the craniotomy and slowly lowered to -1 mm above the skull. The microdrive was grounded to skull screw above the cerebellum. Petroleum jelly (Vaseline, 100% white petrolatum) was gently applied on the cranial window without touching the tetrode, which protected both the brain and microdrive. Next, we further lowered the microdrive to reach the target depth (-300 μm in the cortex). Finally, microdrive was cemented on the skull with dental cements, first with a metabond (Parkell, C&B Metabond Quick Adhesive Cement System, # SKU:S380) followed by an additional layer with dental cement (Steolting, # 51459).

**Tetrode adjustments.** We slowly lowered tetrodes to CA1 stratum pyramidale while the animal was contained in a small, habituated box (dimension in cm; 30 L × 30 W × 30 H) over the course of 3–5 days after microdrive implantation.

**Data acquisition and tetrode placement verification.** We acquired the data using a 32-channel Digital Lynx SX acquisition system (Neuralynx), sampled signals at 32 KHz. We placed the stainless-steel electrodes to screws affixed on the skull above the cerebellum as a ground and tetrodes in the corpus callosum as a reference. A pair of red/green light emitting diodes affixed to the microdrive allowed us to track the position and head direction of animals. At the conclusion of the experiment, we anesthetized mice with avertin and electrode positions were marked by electrolytic lesioning of brain tissue (with 50-μA current for 10–12 s through each tetrode individually). Mice were transcardially perfused with phosphate-buffered saline followed by 4% PFA (Wako chemicals), after which brains were removed and post-fixed for a further 24 h in 4% PFA. Coronal slices 40 μm thick were prepared on a vibratome (Leica), stained with DAPI (Vectashield) and signals were imaged by epifluroscence microscopy (DM6000B, Leica) to confirm electrode placement and reporter expression when appropriate.

**Data segmentation.** The raw LFP and multiunit activity (MUA) data was inspected for quality control measures and channel selection, and the data was split into segments of interest in Neuraview (Neuralynx). We kept the total data length range comparable between genotypes.

**Multiunit spikes.** MUA was extracted by examining waveforms and applying a spike threshold during data acquisition in the Cheetah program (Neuralynx). Briefly, our criteria to define a spike from each tetrode was based on a maximum (typically between 0.2 and 1.0 mV) and minimum (between 0.04 and 0.1 mV) amplitude waveform threshold and other waveform features. This was done during data acquisition using Cheetah software as described previously[26,41,69]. In the CA1-TeTX mice, we set the amplitude threshold such that the presence of high-amplitude high-frequency activity (larger than 1 mV) was not erroneously included as putative spikes; however, the remaining putative spikes that occurred during HFEs/ripples, which were greater in number than in the absence of HFEs/ripples, contaminated normally recorded spike clusters, thus interfering with off-line spike sorting (SpikeSort 3D) of putative single-units. We attempted to eliminate additional spikes during HFEs/ripples with amplitude adjustments to examine clusters of spikes that occur otherwise and were separable; however, we were unsuccessful in our attempts. Therefore, we performed multiunit spike analysis. The firing rate was calculated by calculating by dividing the number of spikes in the MUA by elapsed time, expressed in hertz (Hz), and then normalized to the control group (Supplementary Fig. 3b). For ripples and HFEs associated firing rates (Figs. 2j and 4g), we first detected these events (start and end time indices), calculated the mean firing rate for these durations, and then normalized accordingly. For the autocorrelogram, we

computed the autocorrelation function of the multiunit spike trains for each tetrode with lags, as mentioned in the respective figure panels (Fig. 4h and Supplementary Fig. 3c).

**Power spectral analyses (PSD).** Power spectral analyses with 0.5 or 1 Hz frequency resolution were performed using the pwelch function in MATLAB using a 500 ms time window with a 50% overlap. To clarify trends in PSD, plotted results (Figs. 2d and 3b) were smoothed with Gaussian kernels for plotting purposes only.

For NREM and REM state dependent PSD analysis from quiescence session data, we first examined animal movement velocity (using Neuralynx video tracker data) and extracted the epochs that have no movement. A delta (0.5–4 Hz) to theta (4.5–12 Hz) ratio was computed[41], and putative NREM and REM were identified. A further visual inspection was performed and then we segmented the data into NREM and REM epochs. An equal amount/duration of data was used from both control and CA1-TeTX mice for PSD analysis (1 Hz resolution). We averaged across all tetrodes per mouse before grouping, and the PSD plots shown in Supplementary Fig. 2d were not smoothed.

**Ripples.** The ripple events, present in the recorded LFPs, were detected based on the methods described previously[41]. In order to cover a potential increase in the intrinsic frequency of the SPW-Rs events after CA1 output ablation, the wide band LFP signal was filtered using a band-pass, 69 order, Kaiser window, FIR, zero-phase shift filter with cut off frequencies 80 and 400 Hz. Then, the absolute value of the Hilbert transform was calculated and smoothed with a 50 ms Gaussian window. The putative ripple events were detected as periods when the LFP magnitude exceeded 4 SD (standard deviations) from the mean for more than 30 ms. The initiation and the termination timepoints of each event were defined as the timepoints where the magnitude returned back to mean. Except for the LFP criteria, the MUA, recorded from the same tetrode, was used to calculated smoothed instantaneous firing rate. Firing bursts were then detected, using the same thresholds with the LFP, and any ripple event that did not coincide with MUA bursts was excluded from the subsequent analysis.

The ripple intrinsic frequency was calculated by identifying the frequency corresponding to the maximal power present in the PSD of each one of the ripple waveforms. The peak magnitude was carried out as the maximum of the absolute value of the ripple waveform. The duration of each ripple event was calculated as the total number of points for each event, multiplied by the sampling period (1/1000 s). The total power of each ripple waveform was calculated using the fft function of MATLAB and Equation below.

$$P_{tot(dB)} = 10 \cdot \log\left(\frac{\sum_{n=1}^{N_{FFT}} X(n).X(n)^*}{N_{FFT}}\right)$$

where $X(n)$ represents the Fast Fourier Transform (FFT) of each ripple waveform $(x(n))$ and $N_{FFT}$ the total number of points for the FFT calculation defined as $N_{FFT} = 2^{nextpow2(length(x))}$. The nextpow2 function calculates the nearest power of 2 of its argument.

We averaged across three to five tetrodes from 30 min sleep and 10–12 min of linear track data per mouse to generate the mean waveform and magnitude scalogram of ripples and HFEs shown in Supplementary Fig. 4.

**High-frequency events during linear track exploration.** The high-frequency oscillations (HFEs) during the linear track task were detected only in specific CA1 LFP channels. The channels were chosen after application of continuous wavelet transform (cwt MATLAB function) in the raw LFP data using a complex Morlet wavelet with centre frequency ($f_c$) equal to 1.5 Hz and positive bandwidth parameter ($f_b$) 1 Hz. Dyadic

scales were used based on the equation below.

$$\text{Scales} = 2^i \text{ with } i = nextpow2(c_1): 0.1: nextpow2(c_2)$$

where the pseudo-frequencies corresponding to the scales $s_1 = 2^{nextpow2(c1)}$ and $s_1 = 2^{nextpow2(c2)}$ were equal to 400 Hz and 0.0977 Hz for down sampled signals with sampling frequency $F_s = 1600$ Hz. The wavelet scalogram was then calculated and the LFP channels in which events with frequency components greater than 100 Hz were present, were used as an input to the HFE detection algorithm.

The event detection of these high-frequency oscillations followed the same algorithm with the ripple detection using this time a filter with cut off frequencies 80 and 400 Hz, LFP magnitude threshold at 4 SD and MUA threshold at 2 SD. After the application of the algorithm the timestamps of the beginning, peak magnitude and end of each event were obtained. In order to investigate the spatial properties of the detected high-frequency events, the timestamps of the peak magnitude were used. We hypothesized an analogy between the occurrence of a spike and the occurrence of a high-frequency event represented by the peak magnitude timestamp. Thus, each LFP channel timestamps sequence of events was treated as a "spike sequence" of a "firing cell". By using this assumption, we were able to construct "place field maps". The artificial "firing rate" was calculated for each LFP channel as the number of events per time unit. Additionally, the spatial information measure (bits/spike), typically used to characterize place cells activity, was also carried out based on "firing rate"/occupancy curves. Assuming that the linear track was binned in i spatial bins, the spatial information was calculated as:

$$\text{Spatial information(bits/spike)} = \sum_i P_{spike_i} \cdot \log_2\left(\frac{P_{spike_i}}{P_{occ_i}}\right)$$

where $P_{spikei}$ is the probability of spiking in the spatial bin i, $P_{occi}$ is the occupancy probability in bin i and the summation goes over all the spatial bins. In order to test the significance of the events place properties, 3 different shuffling methods were implemented.

The first method was adapted from[72], where it had been used for place cell analysis. Thus, for each LFP channel the timestamps of the high-frequency events were shifted in time by a random (normally distributed) value between 10 s and the duration of the whole session minus 10 s (the timestamp of the end of each session was wrapped into the timestamp of the beginning). The procedure was repeated 1000 times and the spatial information measure was calculated for each shifted sequence. A high-frequency event sequence was characterised as having place properties only if the spatial information of the original sequence exceeded the 95% percentile of its shuffled distribution.

In the second method, for each sequence, we calculated its inter-event intervals, and we approximated their distribution using the Kernel Density Estimator of MATLAB. Afterwards, for every original sequence we reconstructed 1000 shuffled sequences by drawing random (normally distributed) inter-event intervals from the estimated distribution, keeping the first event timestamp always constant and wrapping the end of each session to the beginning. The spatial information measure was calculated for each shuffled sequence. Again, a high-frequency event sequence was characterised as having place properties only if the spatial information of the original sequence exceeded the 95% percentile of its shuffled distribution.

In the third method we firstly transformed the event sequence to a sequence of inter-event intervals. Then, we randomly chose an index of that sequence as a pivot for swapping the intervals around it. Based on the new inter-event sequence, we then constructed a new shuffled event sequence. We repeated this 1000 times for each one of the original sequences and we calculated the spatial information index for each shuffled sequence. Again, a high-frequency event sequence was characterized as having place properties only if the spatial information

of the original sequence exceeded the 95% percentile of its shuffled distribution.

Spatially triggered HFEs on a lap-by-lap basis on the linear track 2 were plotted (Supplementary Fig. 6a) following the method described previously[66].

The relationship between oscillations and animal movement velocity was examined following the methods as described previously[45].

**Theta phase-locking of the high-frequency oscillations.** The phase relationship between the LFP $\theta$ cycles and the timing of the high-frequency events was calculated using the same techniques described previously[73], for pyramidal cell spikes sequences. Firstly, the LFP traces were filtered in the $\theta$ band (4–12 Hz). Then, the instantaneous $\theta$ phase was calculated using the Hilbert transform. The peaks and troughs of the $\theta$ cycles were assigned to 0° and 180°, respectively. The $\theta$ phase of each event was calculated using interpolation between the previous values, a method that is insensitive to $\theta$ asymmetry. The statistical significance of the phase locking, the preferred occurrence phase and the strength of modulation values was tested using functions of the Circular Statistics Toolbox[74].

**Cumulative co-occurrence Index (CCI).** The Cumulative Co-occurrence Index (CCI) is an objective measure of inter-channel occurrences of high-frequency events in multichannel LFP recordings. An arbitrary high-frequency event $HFE_{mn}$, $m = 1,2,3,\ldots\ldots,M$ & $n = 1,2,3\ldots\ldots,N$ in a channel m of the LFP recording with M channels, is considered to be co-occurring with a high-frequency event $HFE_{pq}$, $p = 1,2,3,\ldots\ldots,M$ & $q = 1,2,3\ldots\ldots,Q$ in another channel p given $p \neq m$ with a unit joint probability $P(HFE_{mn},HFE_{pq}) = 1$, if the time elapsed between the events $T_{mn} - T_{pq} \leq 3$ milliseconds. A threshold of 3 milliseconds accounts for the transmission delay across the hippocampal regions. The variables 'n' and 'q' denote the indices of the arbitrary high-frequency events and the variables 'N' and 'Q' represent the total number of events respectively in m and p channels. $T_{mn}$ and $T_{pq}$ represent the time instant corresponding to the peaks of the high-frequency events, $HFE_{mn}$ and $HFE_{pq}$, respectively. The probability $P_{mn}$ that $n^{th}$ high-frequency event $HFE_{mn}$ in the channel m may co-occur with at least one event each in all other channels is,

$$P_{mn} = \frac{1}{M-1}\sum_{t=1}^{M} P_{tn} \text{ given } t \neq m \text{ \& } m = 1,2,3,\ldots\ldots M \quad (1)$$

The CCI is computed from the probabilities of occurrences of high-frequency events in all channels. The expressions for computing the mean probability $P_m$ that represents the possibility of high-frequency events in channel m coinciding with the events in other channels and the CCI are,

$$CCI = \frac{1}{M}\sum_{m=1}^{M} P_m \text{ Given } P_m = \sum_{n=1}^{N} P_{mn}, m = 1,2,3,\ldots\ldots M \quad (2)$$

**Probability of LFP power distribution during ripples and HFEs in the hippocampus.** For each tetrode, ripple events during the rest sessions and HFEs during linear track exploration were detected as periods when the LFP magnitude exceeded 4 standard deviations from the mean for more than 30 ms. The initiation and the termination timepoints of each event were determined as the timepoints where the magnitude returned back to LFP mean. Subsequently, for each ripple and HFE detected on any tetrode (hub tt), the total LFP power was calculated using the fft function of MATLAB from all 5 tetrodes. The powers in the hub tt and the concurrent LFP segments in other tts were normalized by the cumulative concurrent LFP power. Thus, the normalized power of ripples/HFEs LFP segment reflects the probability of the power being concentrated on that tt (i.e., that region of CA1) during the high-frequency ripple event. Averaging the normalized power or regional

power concentration probability for all high-frequency events in the hub tts reflects the overall probability of the power distribution across the hippocampus during the high-frequency events (Fig. 4d).

**Cross power spectral density.** Relative phases between LFP recordings from different tetrodes/channels are computed from their cross power spectral density (CPSD) estimate. The CPSD is estimated via Welch's averaged, modified periodogram method of spectral estimation. Both input signal arrays comprising N numbers of samples after detrending are divided into eight segments each with $\lfloor N/4.5 \rfloor$ overlapping samples with the adjoining segments. The segments are subjected to a Hamming window function to avoid spectral leakage. The number of DFT points in the CPSD is restricted to $10^5$ for esthetic visualization. The whole-length (-10 min) linear-track LFP recordings are used in the analysis. The relative phase is computed from the four-quadrant inverse tangent function of the complex CPSD representation. In general, the relative phase $\varphi$, and the CPSD $P_{xy}$ of two stationary random processes $x_n$ and $y_n$ are computed from their cross-correlation sequence $R_{xy}$ as,

$$\varphi(\omega) = \angle P_{xy}(\omega), P_{xy}(\omega) = \sum_{m=-\infty}^{+\infty} R_{xy}(m)e^{-j\omega n} \quad (3)$$

where cross-correlation sequence is,

$$R_{xy}(m) = E(x_{n+m}y_n^*) = E(x_n y_{n-m}^*) \quad (4)$$

In Eq. (4), E(.) denotes the expectation operator.

**Magnitude-squared coherence.** The magnitude-squared (MS) coherence estimate is used to analyze the correspondence of spectral attributes between various possible combinations of LFP recordings from different tetrodes/channels. The MS coherence estimate reflects correspondence of spectral power distribution at same frequencies in the LFP recordings from two tetrodes. In general, the MS coherence between two signals x and y is a function of the respective power spectral density estimates, $P_{xx}(f)$ and $P_{yy}(f)$, and their cross power spectral density estimate, $P_{xy}(f)$ as,

$$C_{xy}(f) = \frac{|P_{xy}(f)|^2}{P_{xx}(f)P_{yy}(f)} + \quad (5)$$

The magnitude-squared coherence estimate will be equal to one when the spectral power at a particular frequency is equal in both the input signals and reduces in proportion to the difference in spectral power values. Both input signal arrays comprising N numbers of samples after detrending are divided into eight segments each with $\lfloor N/4.5 \rfloor$ overlapping samples with the adjoining segments. The segments are subjected to a Hamming window function to avoid spectral leakage. The number of the DFT points in the power spectral density estimates is restricted to $10^5$ for esthetic visualization.

### Quantification and statistical analysis

Statistics were performed in GraphPad Prism (version 9.5.0) or MATLAB (versions 2016a, 2019a and 2022b). Statistical significance was calculated as noted in the appropriate figure legends and text. Briefly, to test whether the means of paired observations were significantly different, we used a paired $t$ test. To test whether mean values differed significantly between the two groups, we used independent samples t-test or Mann-Whitney U based on the distribution of observations. Two-way ANOVA or two-way repeated measures ANOVA were used when two or more variables were compared. All tests were corrected for multiple comparisons between frequencies using a procedure that controls the false discovery rate under arbitrary

dependence assumptions (i.e., two-stage step-up method of Benjamini, Krieger, and Yekutieli). Statistical significance was set at 0.05.

## Reporting summary

Further information on research design is available in the Nature Portfolio Reporting Summary linked to this article.

## Data availability

This study did not generate new unique reagents. All data necessary to assess the conclusions of this research are available in the Source Data, text and supplementary information. Any additional information required to reanalyze the data reported in this paper is available from the lead contact upon request. Source data are provided with this paper.

## Code availability

The study made use of previously published code[26] (https://www.nature.com/articles/nn.4311), as well as the publicly available Chronux toolbox (version 2.12.v03; http://chronux.org/) and MATLAB File Exchange: Circular Statistics Toolbox (https://www.mathworks.com/matlabcentral/fileexchange/10676-circular-statistics-toolbox-directional-statistics). In addition, we developed a MATLAB function to analyze the Cumulative Co-occurrence Index (CCI), which can be found along with an instructions and demo file and necessary data at: https://github.com/ChinnakkaruppanAdaikkan/Cumulative-Cooccurance-Index.

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

## Acknowledgements

We thank all members of the McHugh lab for discussions and feedback on the manuscript and E. McNamara for mouse colony maintenance. This work was supported by Grant-in-Aid for Scientific Research from MEXT (19H05646, 23H05478; T.J.M.), RIKEN Center for Brain Science (T.J.M.), the MIT Picower Institute Innovation Fund (L.-H.T.), and CBR startup fund (C.A.), and DBT/Wellcome Trust Intermediate Fellow Grant (IA/I/22/1/506257; C.A.).

## Author contributions

Conceptualization: C.A. and T.J.M.; methodology: C.A., J.J., J.W., D.P., S.J.M., R.B., A.J.Y.H., and G.F.; investigation: C.A., J.J., J.W., D.P., and G.F.; writing—original draft: C.A. and T.J.M.; writing—review and editing: C.A., T.J.M., G.F., D.P., S.J.M., R.B., and A.J.Y.H.; funding acquisition: T.J.M., L.-H.T., and C.A.; resources, L.-H.T. and T.J.M.; supervision: C.A. and T.J.M.

## Competing interests
The authors declare no competing interests.
