## [Peer Review File · Nature Communications]

REVIEWER COMMENTS

Reviewer #1 (Remarks to the Author):

In this manuscript, the authors use conditional genetic manipulations with tetanus toxin (TTX) blockage of CA1 pyramidal cell's synaptic transmission to evaluate the role of feedback inhibition, which is heavily impaired by the manipulation, in various aspects of hippocampal dynamics, including ripple oscillations during quiescence and spatially selective ensemble dynamics during movement. The findings are clearly presented, the results from the different experiments are consistent and the discussion is well presented in line with the literature. I find the study well conducted overall, the data clearly presented and the conclusions convincing. I have some comments that the authors could consider to add/address to their current manuscript:

Minor comments:

-The power spectrum in the control data (shown in 2d and more clearly in 3b) does not show power density around 150Hz (ripple frequency) other than the regular decay as a power law. A typical frequency spectrum would show some high power due to the repeated occurrence of network events at this frequency. Is this not separating by brain states (theta, non-theta, sleep...)? It could be helpful to separate it and show it per state so is clear that theta and ripple power are present, as they are clearly shown in the raw traces shown in the LFP.

-Some of the authors showed similar results in an earlier paper, Boehringer et al, with the same manipulation but in CA2 pyramidal cells. Blocking CA2 affected inhibition in CA3, and blocking CA1 affected inhibition in CA1. But given similar results regarding the population spikes during theta that remain spatially selective, is it possible that this type of manipulation result in a general E/I imbalance in the whole hippocampus independently of the region of manipulation?

-Following up on the previous point, providing some info regarding other hippocampal subregions could be helpful. The authors clarify that there is no access to single units due to the saturation of the signal but perhaps using other read outs (gamma or LFP profiles if available) could help.

-HFEs show clear spatial selectivity but what about the number of "place cells" in each condition? The authors could provide some numbers on how many place cells are in each direction during the control sessions versus how many spatially selective HFEs in the TTX condition (or similar) as well as spatial information for each, number of fields...

Reviewer #2 (Remarks to the Author):

The authors of this study aimed to reduce or eliminate the feedback excitation in the CA1 region of the hippocampus to reveal its contribution to the oscillatory activities generated in behaving mice. To this end, tetanus toxin light chain was expressed in CA1 region of CaMKIIa-Cre mice in a Cre dependent manner, an intervention that is known to suppress transmitter release of the infected neurons. However, this approach may not be selective for CA1 pyramidal cells as a recent paper indicates (Veres et al., 2023). The authors of this paper demonstrated that many interneuron types can be infected by using

CaMKIIa promoter even though the presence of the genetic constructs cannot be detected using immunostaining. Thus, using CaMKIIa-Cre one will affect the function of both CA1 pyramidal cells and a portion of GABAergic interneurons. This disadvantage of the applied method precludes drawing solid conclusion about the contribution of feedback excitation to oscillogenesis which is the main aim of the present MS. No attempt has been made in this MS to clarify the impact of the intervention on GABAergic transmission. The authors should have performed the analysis of mIPSCs recorded in CA1 pyramidal cells and studied the features of evoked IPSCs as they did for eEPSCs shown in Fig 1e. Without these experiments, it is hard to accept that the used approach was selective for CA1 pyramidal cells.

Minor:

1. The authors claim that gamma oscillations in CA1 should be driven in a feedforward manner based on their own results. However, previous studies (e.g., Csicsvari et al., 2003, Zemankovics et al., 2013) have clearly demonstrated that the slow gamma in the CA1 is driven by the CA3.
2. "oscillations induced by the cholinergic receptor agonist carbachol require exclusively glutamatergic neurotransmission". This is misinterpretation of the results of the cited studies. Fisahn et al., 1998 and others clearly showed that cholinergic model of gamma oscillations depends on both glutamatergic and GABAergic synaptic transmission.

Reviewer #3 (Remarks to the Author):

In this study, Adaikkan and colleagues use Cre-dependent expression of tetanus toxin to specifically suppress synaptic transmission from CA1 pyramidal cells (and thereby feedback inhibition) and investigate the contribution of feedback inhibition to CA1 network dynamics. A series of immunohistochemistry and slice experiments are first carried to confirm that TeTX expression is selective to CA1 PCs, that synaptic transmission from CA1 PCs is reduced (reduction of subicular EPSCs from CA1 stimulation), and that feedback inhibition is reduced in CA1 (reduction of spontaneous IPSC frequency, increased population spike and paired pulse ratio). The impact of suppressing feedback inhibition is then analyzed in terms of ripple, theta and gamma oscillations in the behaving mouse. Ripple power, frequency, duration and cell spiking during ripple are increased while the time interval between ripples is unchanged. Theta power (but not frequency) is increased while theta modulation of low/high gamma is relatively unaffected. Furthermore, high-frequency oscillation events (HFE) somewhat similar to ripples are observed in TeTX mice, phase-locked to the trough of theta oscillations and relatively localized both in space on a linear track and across the CA1 area.

The manuscript is generally well written and constitutes the first study that investigates the contribution of feedback inhibition to CA1 network dynamics. Sophisticated tools are used and the suppression of feedback inhibition is carefully tested in vitro. It is only unfortunate that individual neurons could not be isolated due to voltage saturation, which would have allowed for more in depth analysis of the temporal organization of cell population.

Specific comments:

_ A high magnification display of a HFE would be useful in order to get a visual comparison with ripples.

_ To make the point that HFEs occur at unique/stable locations on the linear track across the session (Line 243), place field maps of HFE need to be shown for several consecutive laps of mouse running on the linear track and the stability should be quantified for instance via correlation of place field maps.

_ Line 178: "...both low (30 – 50 Hz) and high (50 – 100 Hz) gamma power were comparable across the groups (Fig. 3b; Supplementary Fig. 3a)."

There are no separate analysis for the power of low and high gamma in the figures mentioned.

^[L]_{SEP} Line 202: "Indeed, we observed high-frequency oscillatory events (HFEs), present only in CA1-TeTX mice, occurring at an average frequency of 23.24 ± 7.62 events/min (Fig. 3e)."

In figure 4, occurrence and theta phase of HFEs are also reported for control mice. Were there HFEs in control mice during theta?

_ Line 219: "Thus, we examined the relationship between the mean peak amplitudes of HFEs detected in any given electrode and the corresponding peak amplitudes of all other channels. The mean peak phase and amplitude correlations across CA1 locations during HFEs were smaller during active exploration than the quiescence session in CA1-TeTX mice (Fig. 4d- 4f; Supplementary Fig. 4a- 4c)."^[L]_{SEP}

Fig. 4d-4f and Supplementary Fig 4a-4c (and figure legend) do not match with what is described in the text.

_ Suppl Fig4d legend: "...from 7 CA1-TeTX mice"

The figure shows more than 7 plots. I guess this is because there can be more than one electrode per mouse. Please either specify in the legend "x to x electrode per mouse" or add on each plot a label for mouse and electrode number.

_ Line 287: "One unfortunate consequence of the size and frequency of the pathophysiological events in the CA1-TeTX mice, both during rest and movement, was the saturation of conventional amplification, precluding the recording of single unit activity. Thus, while we could examine multiunit spiking outside of the isolated HFEs and ripples, it was impossible to adjust the amplification to a level that allowed the clustering of single neurons."

It seems that spike rate was measured during HFEs for Fig4g,h,i. How this was possible?^[L]_{SEP}

_ Suppl Fig. 3c: s instead of ms for the x axis label

_ Line 244: "...resembling the activity typically seen in single place cells in CA1 (Fig. 4j; Supplementary Fig. 4d)."^[L]_{SEP}

Fig4k instead of 4j.

Point-by-Point Response

Reviewer #1

In this manuscript, the authors use conditional genetic manipulations with tetanus toxin (TTX) blockage of CA1 pyramidal cell's synaptic transmission to evaluate the role of feedback inhibition, which is heavily impaired by the manipulation, in various aspects of hippocampal dynamics, including ripple oscillations during quiescence and spatially selective ensemble dynamics during movement. The findings are clearly presented, the results from the different experiments are consistent and the discussion is well presented in line with the literature. I find the study well conducted overall, the data clearly presented and the conclusions convincing. I have some comments that the authors could consider to add/address to their current manuscript:

We thank the reviewer for their support and suggestions; please find our modifications detailed below.

Minor comments:

-The power spectrum in the control data (shown in 2d and more clearly in 3b) does not show power density around 150Hz (ripple frequency) other than the regular decay as a power law. A typical frequency spectrum would show some high power due to the repeated occurrence of network events at this frequency. Is this not separating by brain states (theta, non-theta, sleep...)? It could be helpful to separate it and show it per state so is clear that theta and ripple power are present, as they are clearly shown in the raw traces shown in the LFP.

Fig 3b shows the power profile from active exploration (i.e., linear track) data, and thus, we do not expect any power in the ripple band in control mice. Fig 2d is from the rest/quiescence session and we detected ripples with the mean frequency of 138.8 ± 3.58 Hz in control mice. Accordingly, we see a modest increase in power density after 100 Hz in Fig 2d; however, we agree with the reviewer that it does not appear robust. This is likely because ripples predominantly occur during slow-wave (NREM) sleep/quiet immobility, but these data included NREM, REM, and awake periods. Per the reviewer's suggestion, we segmented the data into putative NREM & REM sleep based on movement and theta/delta ratio and examined the power (see Methods, page 11, lines 505-511), and as expected, we indeed found higher power in the expected ripple frequency (see excerpt from new Supplemental Figure 2d below in Response Figure 1).

Response Fig 1. Control mice LFP power spectrum from NREM sleep periods.

We have included these new analyses in the revised manuscript (Supplementary Fig. 2d) (described on page 4, lines 159-161).

-Some of the authors showed similar results in an earlier paper, Boehringer et al, with the same manipulation but in CA2 pyramidal cells. Blocking CA2 affected inhibition in CA3, and blocking CA1 affected inhibition in CA1. But given similar results regarding the population spikes during theta that remain spatially selective, is it possible that this type of manipulation result in a general E/I imbalance in the whole hippocampus independently of the region of manipulation?

We did touch on this in the discussion in the original submission, but have expanded this section in the revised manuscript (please see page 6, lines 289-292).

We do not think TeTX expression in CA1 caused a general E/I imbalance in the whole hippocampus; three specific observations, detailed below, support this conclusion:

1. During active exploration, slow gamma, known to depend on CA3 input, is unaltered in these mice, strongly suggesting intact CA3 function.
2. In select mice we had occasional tetrodes targeting CA1 in regions in which TeTX expression levels were undetectable due to focal expression of the viral construct (see the arrow in the B panel of the figure below). Although they were positioned in the same hemisphere (typically <500 μ m away) in which we observed the altered activity on other tetrodes due to TeTX expression, the LFP recorded on these tetrodes showed ripple power spectrums comparable to control mice (lower right wavelet scalogram in Response Figure 2 below). As these data are limited, we did not include them in the manuscript, however we prepared the figure below for the referee's reference.

Response Fig 2. **A.** Experimental diagram, ipsilateral recordings were made both inside and outside areas of TeTX expression in the same mouse. **B.** Representative single-plane confocal images show expression of mCherry in an example mouse, white arrowhead indicates location of tetrode located outside area of TeTX expression. **C, D** Wavelet scalogram of LFP power during ripple form control (**C**) and CA1-TeTX (**D**) mouse. Top panels show example from a tetrode inside the area of expression, middle panels from outside the expression area. Bottom panels show multiunit spiking activity during the examples above, blue line- tetrode outside the area of expression, orange line- tetrode inside area of expression.

3. In addition, we observed that ripples occurred selectively during NREM sleep as expected & we did not detect high-frequency oscillatory activity (i.e., ripples) during REM-theta sleep (Supplementary Fig. 2d) in the TeTX mice.

-Following up on the previous point, providing some info regarding other hippocampal subregions could be helpful. The authors clarify that there is no access to single units due to the saturation of the signal but perhaps using other read outs (gamma or LFP profiles if available) could help.

Thank you for this suggestion. Although we limited our recordings to the CA1 region in this study, we interpret the fact that slow gamma, known to depend on CA3 input, is unaltered in these mice as a strong indication of intact CA3 function. We explicitly mention this in the revised discussion.

-HFEs show clear spatial selectivity but what about the number of “place cells” in each condition? The authors could provide some numbers on how many place cells are in each direction during the control sessions versus how many spatially selective HFEs in the TTX condition (or similar) as well as spatial information for each, number of fields...

We appreciate the reviewer’s suggestion. As described in the manuscript, we could not reliably isolate single units in TeTX mice (page 11, lines 487-497), preventing place cell analysis. However, we now provided additional information regarding how many spatially selective tetrodes were observed in each TeTX mouse. All mice showed spatially selective tetrodes and a total of 27 spatially selective tetrodes (TeTX1 = 5 tts (tetrodes), TeTX2 = 3 tts, TeTX3 = 5 tts, TeTX4 = 4 tts, TeTX5 = 4 tts, TeTX6 = 3 tts, TeTX7 = 3 tts) with one or more place field like patterns were observed per tetrode (Supplementary Fig. 5d).

Reviewer #2 (Remarks to the Author):

The authors of this study aimed to reduce or eliminate the feedback excitation in the CA1 region of the hippocampus to reveal its contribution to the oscillatory activities generated in behaving mice. To this end, tetanus toxin light chain was expressed in CA1 region of CaMKIIa-Cre mice in a Cre dependent manner, an intervention that is known to suppress transmitter release of the infected neurons. However, this approach may not be selective for CA1 pyramidal cells as a recent paper indicates (Veres et al., 2023). The authors of this paper demonstrated that many interneuron types can be infected by using CaMKIIa promoter even though the presence of the genetic constructs cannot be detected using immunostaining. Thus, using CaMKIIa-Cre one will affect the function of both CA1 pyramidal cells and a portion of GABAergic interneurons. This disadvantage of the applied method precludes drawing solid conclusion about the contribution of feedback excitation to oscillogenesis which is the main aim of the present MS. No attempt has been made in this MS to clarify the impact of the intervention on GABAergic transmission. The authors should have performed the analysis of mIPSCs recorded in CA1 pyramidal cells and studied the features of evoked IPSCs as they did for eEPSCs shown in Fig 1e. Without these experiments, it is hard to accept that the used approach was selective for CA1 pyramidal cells.

We thank the reviewer for raising an important point, and we agree that no genetic tool is perfect and care must be taken to correctly interpret the results of cell-type specific circuit interventions. Moreover, the exact approach employed and the details of the experimental protocol are key factors in achieving maximum specificity & interpretability. We address these issues as it pertains to our experiments below:

1. The Veres et al. paper is interesting; however, as detailed below, it primarily finds CamKII promoter driven expression in interneurons when a small promoter fragment is used in an AAV virus, in contrast to a transgenic line with CaMKII as the promoter. The mouse line (Tg(Camk2a-cre)CW2Stl, MGI: 3530562) we use here was first published in 2001 (Zeng et al, Cell 2001) and has been employed in at least a dozen studies to specifically target genetic manipulations to the pyramidal cells in CA1, with no reported evidence of expression in any interneurons. It contains an 8.5kb fragment of the CamKII promoter, in contrast with the minimal 1.3kb promoter typically employed in AAV packaging.
2. Moreover, the interneuron expression Veres et al. observed in their experiments was maximal in the BLA and, to a lesser extent, the PFC, with *much* lower numbers of labeled interneurons in the CA1 region (please see their Table 2). Thus, they find that in contrast to the PFC and BLA, in CA1, the expression in interneurons is drastically lower, particularly in the CamKII-GFP transgenic strain, which is the best approximation of the method we employ here. Furthermore, as has been evident in the field and highlighted in the Veres et al. study, numerous other factors can influence the expression of a given viral payload, including the AAV serotype, the titer of the virus (note that when Veres et al. reduced the titer of their CamKII-cre virus 10x they observed absolutely no interneuron expression in CA1), and the size and sequence of the promoter fragment used (as the reviewer might know, the limited size of the AAV genome requires a minimum promoter sequence).

Thus, the methods employed differ considerably between our and Veres et al. study. Importantly, as we demonstrate in Figure 1a, b in our manuscript, we find zero overlap between GAD67, a broad marker of inhibitory neurons, and Cre-mediated viral expression, both in mice injected with the TeTX expressing virus, as well as in mice injected with a mCherry control virus. As a positive control, we conducted

further analyses with this same GAD67 antibody in brain slices from PV-Cre mice and SOM-Cre mice injected with the TeTX.mCherry virus. In both lines we observe complete overlap of the GAD67 signal with the reporter mCherry expression (Response Figure 3 below, single plane confocal images). These observations confirm that GAD67 is a good marker for interneurons and further illustrate that if TeTX.mCherry was expressed in interneurons in CA1-TeTX mice, we would likely detect it with immunostaining.

Response Fig 3. GAD67 immunostaining detects viral expression in PV-Cre and SOM-Cre mice

Regarding evoked IPSCs, it is unclear how we would be expected to perform or analyze the evoked IPSCs analysis the reviewer suggested in our experiments, especially if the “off-target” expression in GABAergic neurons is relatively sparse. Further, technically, it is difficult to specifically stimulate specific GABAergic neurons (without labeling them via optogenetic channels) in CaMKII-Cre: AAV.TeTX background. Importantly, when designing the IPSC analysis in Figure 1, we specifically chose to record spontaneous IPSCs, as they include both spike-dependent and independent GABAergic transmission, in contrast to mIPSC, which are used to study GABAergic transmission in the absence of spikes.

Overall, given that the titer & the volume of AAV infused in multiple lines of mice in our hands selectively label Cre-expressing neurons, and further that no GABAergic markers are found in CA1.TeTX.mCherry+ cells, we hope now that the reviewer is convinced that TeTX was expressed specifically in CA1 pyramidal neurons in these experiments.

Minor:

1. The authors claim that gamma oscillations in CA1 should be driven in a feedforward manner based on their own results. However, previous studies (e.g., Csicsvari et al., 2003, Zemankovics et al., 2013) have clearly demonstrated that the slow gamma in the CA1 is driven by the CA3.

The referee is correct, and we completely agree that Csicsvari and others have shown that slow gamma in CA1 is driven by CA3 (i.e. feedforward refers to the fact the oscillation is a result of feedforward input from CA3 to CA1). To that point, in the previous manuscript we wrote: “*In contrast, we saw no change in*

the frequency, amplitude or phase-amplitude coupling of both slow and fast gamma oscillations. This observation is consistent with the hypothesis that these faster oscillations present during movement are driven by coupling with oscillations driven by CA3 (slow gamma) and EC (fast gamma)” to reflect this. In the revised manuscript this can be found on page 6, lines 286-289.

2. “oscillations induced by the cholinergic receptor agonist carbachol require exclusively glutamatergic neurotransmission”. This is misinterpretation of the results of the cited studies. Fisahn et al., 1998 and others clearly showed that cholinergic model of gamma oscillations depends on both glutamatergic and GABAergic synaptic transmission.

We apologize for our oversight. We have modified this point in the revised manuscript (page 2, lines 57-59).

Reviewer #3 (Remarks to the Author):

In this study, Adaikkan and colleagues use Cre-dependent expression of tetanus toxin to specifically suppress synaptic transmission from CA1 pyramidal cells (and thereby feedback inhibition) and investigate the contribution of feedback inhibition to CA1 network dynamics. A series of immunohistochemistry and slice experiments are first carried to confirm that TeTX expression is selective to CA1 PCs, that synaptic transmission from CA1 PCs is reduced (reduction of subicular EPSCs from CA1 stimulation), and that feedback inhibition is reduced in CA1 (reduction of spontaneous IPSC frequency, increased population spike and paired pulse ratio). The impact of suppressing feedback inhibition is then analyzed in terms of ripple, theta and gamma oscillations in the behaving mouse. Ripple power, frequency, duration and cell spiking during ripple are increased while the time interval between ripples is unchanged. Theta power (but not frequency) is increased while theta modulation of low/high gamma is relatively unaffected. Furthermore, high-frequency oscillation events (HFE) somewhat similar to ripples are observed in TeTX mice, phase-locked to the trough of theta oscillations and relatively localized both in space on a linear track and across the CA1 area.

The manuscript is generally well written and constitutes the first study that investigates the contribution of feedback inhibition to CA1 network dynamics. Sophisticated tools are used and the suppression of feedback inhibition is carefully tested in vitro. It is only unfortunate that individual neurons could not be isolated due to voltage saturation, which would have allowed for more in depth analysis of the temporal organization of cell population.

We thank the referee for the supportive comments and agree it would have been great to dig deeper into the changes in the temporal organization of CA1 activity in the absence of feedback inhibition.

Specific comments:

_ A high magnification display of a HFE would be useful in order to get a visual comparison with ripples.

Thank you for the suggestion; we have performed this additional analysis and included a new figure in the revised manuscript (Supplementary Fig. 4) that shows the mean wavelet scalogram and waveforms of ripples (rest) and HFEs (linear track exploration) from each of 7 CA1-TeTX mice.

_ To make the point that HFEs occur at unique/stable locations on the linear track across the session (Line 243), place field maps of HFE need to be shown for several consecutive laps of mouse running on the linear track and the stability should be quantified for instance via correlation of place field maps.

Thank you for the suggestion. Per the reviewer's suggestion, we have performed this additional analysis and added this as Supplementary Fig. 6 in the revised manuscript.

_ Line 178: "...both low (30 – 50 Hz) and high (50 – 100 Hz) gamma power were comparable across the groups (Fig. 3b; Supplementary Fig. 3a)."

There are no separate analysis for the power of low and high gamma in the figures mentioned.

Our repeated measures ANOVA followed by post hoc testing did not find a significant difference at any specific frequency across the 30 – 100 Hz range between control and CA1-TeTX mice (see Response Figure 4 below for p values). We thus did not provide a separate analysis. However, we apologize for our error and per the reviewer’s suggestion, we have now added the statistical analysis for the gamma band to the revised Figure 3b and Supplemental Figure 2d.

Response Fig 4. Results of post-hoc comparisons corrections for repeated testing across frequencies

_ Line 202: “Indeed, we observed high-frequency oscillatory events (HFEs), present only in CA1-TeTX mice, occurring at an average frequency of 23.24 ± 7.62 events/min (Fig. 3e).”

In figure 4, occurrence and theta phase of HFEs are also reported for control mice. Were there HFEs in control mice during theta?

We apologize for the confusion. The data in Figures 4c through 4f included recordings from both movement and rest (this was noted in the figure panels and legend). This was intentional so we could use normal ripple events in the controls as a comparison of the co-occurrence of high-frequency oscillations (ripples in controls, ripples and HFEs in the CA1-TeTX mice) across different recording sites. We have clarified this in the text and legends of the revised manuscript (page 5, lines 218-220).

_ Line 219: “Thus, we examined the relationship between the mean peak amplitudes of HFEs detected in any given electrode and the corresponding peak amplitudes of all other channels. The mean peak phase and amplitude correlations across CA1 locations during HFEs were smaller during active exploration than the quiescence session in CA1-TeTX mice (Fig. 4d- 4f; Supplementary Fig. 4a- 4c).”

Fig. 4d-4f and Supplementary Fig 4a-4c (and figure legend) do not match with what is described in the text.

We again apologize for the confusion. Please note that the higher cool colors (blue) in the amplitude correlation map during the active exploration session (right panel, Fig. 4d) indicate lower amplitudes

relative to the diagonal box. Further, a higher relative phase (Fig. 4e) indicates poor phase consistency (i.e., lower correlation between tetrodes).

Nonetheless, although the meanings remain unchanged, we modified the sentence for clarity and brevity. Our revised sentence is as follows: *...The mean peak phase and amplitude correlations across CA1 locations during HFEs significantly differed between active exploration and the quiescence session in CA1-TeTX mice..* (page 5, lines 223-226).

_ Suppl Fig4d legend: "...from 7 CA1-TeTX mice"

The figure shows more than 7 plots. I guess this is because there can be more than one electrode per mouse. Please either specify in the legend "x to x electrode per mouse" or add on each plot a label for mouse and electrode number.

The referee is correct; we have specified this in the figure (Supplementary Fig. 5d) as suggested.

_ Line 287: "One unfortunate consequence of the size and frequency of the pathophysiological events in the CA1-TeTX mice, both during rest and movement, was the saturation of conventional amplification, precluding the recording of single unit activity. Thus, while we could examine multiunit spiking outside of the isolated HFEs and ripples, it was impossible to adjust the amplification to a level that allowed the clustering of single neurons."

It seems that spike rate was measured during HFEs for Fig4g,h,i. How this was possible?^[SEP]

Briefly, our criteria to define a spike from each tetrode is based on a maximum (typically between 0.2 & 1.0 mV) and minimum (between 0.04 & 0.1 mV) amplitude waveform threshold and other waveform features. This was done during data acquisition using Cheetah software (Neuralynx); this method worked reliably in our hands across several projects. In the CA1-TeTX mice, we set the amplitude threshold such that the presence of high-frequency activity with larger amplitudes (larger than 1mV) was not erroneously included as putative spikes; however, the remaining putative spikes that occurred during HFEs/ripples were dramatically higher than in the absence of HFEs/ripples. However, these spikes contaminated other spike clusters & appeared between clusters of spikes, interfering with the offline spike sorting for putative single units. As reported in the original submission, we attempted to eliminate additional spikes during HFEs/ripples with amplitude adjustments to examine those clusters of spikes that occur otherwise and are separable; however, we were unsuccessful in our attempt. Therefore, we performed multiunit spike analysis during HFEs and ripples, as single-unit separation was difficult. We have elaborated this further in the methods of the revised manuscript (page 11, lines 487-501).

_ Suppl Fig. 3c: s instead of ms for the x axis label

Thank you for catching our error, this has been corrected.

_ Line 244: "...resembling the activity typically seen in single place cells in CA1 (Fig. 4j; Supplementary Fig. 4d)."^[SEP]

Fig4k instead of 4j.

Thank you for catching our error, this has been corrected.

REVIEWER COMMENTS

Reviewer #1 (Remarks to the Author):

I think the authors made a good effort in addressing the reviewers concerns and clarified all raised points. I therefore now recommend the paper for publication.

Reviewer #2 (Remarks to the Author):

In the revised version of this study the authors have not made any attempt to clarify whether GABAergic cell function is or is not altered upon expressing TeTX in CaMKII-Cre mice. The authors claimed that they do not see any labeling in PV and SOM interneurons in CA1, therefore interneurons should not be infected by TeTX. However, the study by Veres et al., 2023 showed that interneurons can be infected, and their function altered by CaMKII promoter driven constructs even in those cases when the reporter proteins seem to be not present in their somata (see Fig. 4). Thus, visual inspection of labeling may be misleading and may cause erroneous conclusion, stating that GABA release is unaffected e.g., upon TeTX expression in CaMKII-Cre mice. To convince the readers that indeed the approach used in the study to silence CA1 pyramidal cells is specific, the authors need to perform experiments that directly investigate the function of GABAergic transmission and not only the expression of reporter proteins. For instance, as I proposed earlier, mIPSCs should be recorded and compared as the authors did for mEPSCs in Fig. 1h. The rationale beyond this experiment is that if TeTX silences a portion of GABAergic cells, the rate of mIPSCs recorded in CA1 pyramidal cells should be reduced. In addition, they should record evoked IPSCs in CA1 pyramidal cells as they did for eEPSCs in Fig. 1e. Similarly, the amplitude of pharmacologically isolated eIPSCs recorded in CA1 should be compared as the function of stimulus intensity. If any of these experiments reveal no change in the properties of inhibitory transmission, then the conclusion, stating that the used manipulation only impacts feedback excitation, will be solid. At present, the conclusion of the work is ambiguous and, therefore, it needs further investigation.

Reviewer #3 (Remarks to the Author):

The authors have thoroughly address my comments and I only have a minor remaining comment:

_ Regarding my comment on Fig.4d, I realized that the source of confusion came from the labeling of the color coded calibration bar (“High frequency power during HFEs (dB)”), which still need to be changed to something like “Pearson correlation coefficient” with a range of 0 to 1, as well as the figure legend (something like “Heatmaps show the correlation of high-frequency power between tetrodes during rest and exploration..” instead of “Heatmaps show the LFP power of high-frequency during ...”)

Reviewer #1 (Remarks to the Author):

I think the authors made a good effort in addressing the reviewers concerns and clarified all raised points. I therefore now recommend the paper for publication.

We thank the reviewer for their support.

Reviewer #2 (Remarks to the Author):

In the revised version of this study the authors have not made any attempt to clarify whether GABAergic cell function is or is not altered upon expressing TeTX in CaMKII-Cre mice. The authors claimed that they do not see any labeling in PV and SOM interneurons in CA1, therefore interneurons should not be infected by TeTX. However, the study by Veres et al., 2023 showed that interneurons can be infected, and their function altered by CaMKII promoter driven constructs even in those cases when the reporter proteins seem to be not present in their somata (see Fig. 4). Thus, visual inspection of labeling may be misleading and may cause erroneous conclusion, stating that GABA release is unaffected e.g., upon TeTX expression in CaMKII-Cre mice. To convince the readers that indeed the approach used in the study to silence CA1 pyramidal cells is specific, the authors need to perform experiments that directly investigate the function of GABAergic transmission and not only the expression of reporter proteins. For instance, as I proposed earlier, mIPSCs should be recorded and compared as the authors did for mEPSCs in Fig. 1h. The rationale beyond this experiment is that if TeTX silences a portion of GABAergic cells, the rate of mIPSCs recorded in CA1 pyramidal cells should be reduced. In addition, they should record evoked IPSCs in CA1 pyramidal cells as they did for eEPSCs in Fig. 1e. Similarly, the amplitude of pharmacologically isolated eIPSCs recorded in CA1 should be compared as the function of stimulus intensity. If any of these experiments reveal no change in the properties of inhibitory transmission, then the conclusion, stating that the used manipulation only impacts feedback excitation, will be solid. At present, the conclusion of the work is ambiguous and, therefore, it needs further investigation.

We are in complete agreement with the reviewer's statement that the specificity of the genetic manipulation employed in these experiments is crucial to our interpretation of the data. However, while we appreciate the reviewer's opinions, we believe, as do the other two reviewers, that we have achieved this aim with the current data and we welcome this chance to clarify both why the claim about fluorophore labeling described in Figure 4 of the Veres et al study is patently incorrect in the scope of our experimental design, as well as revisit why we believe the in vitro slice experiment suggested is not feasible.

1. Regarding the point about CaMKII driving expression even when reporter genes are not detectable (Veres et al, Figure 4): We have no reason to doubt the veracity of the Veres et al figure, in which the smaller version of the CaMKII promoter was used to directly drive expression of a fluorophore fused to an optogenetic channel in an AAV construct and reported expression levels at which optogenetic channels were functional, but fluorophore visualization was not achieved. However, in our experimental design, the transgenic mouse employed expressed the Cre recombinase (not a fluorophore and/or a protein designed to alter circuit function) under the control of the 8.5kB CaMKII promoter fragment. The expression of TeTX-mCherry (or mCherry alone in control mice) was thus driven by the virally packaged and very robust EF1a promoter in a Cre-dependent fashion.

Thus, even a small amount of the Cre recombinase being present in an individual neuron (ie. “leak” in interneurons due to the CaMKII promoter) would result in inversion of the expression construct (via the DIO sequence) and robust and detectable expression of the fluorophore due to the EF1a promoter. This is exactly why we included, for the reviewer’s benefit, the figure below in our previous response, which clearly demonstrates that the EF1a driven AAV construct is robustly detectable in GABAergic interneurons in the presence of Cre. Further, the reviewer’s comments again equivocate CaMK2a::Cre mice driven expression and CaMK2a promoter-driven AAV expression - we respectfully disagree with this assessment (mice age, promoter length, and viral titer all matter- these are entirely different between ours and Veres et al).

2. As we elucidated in our previous response regarding evoked IPSC, it is unclear how we would be expected to perform or analyze the evoked IPSC analysis the reviewer suggested in our experiments, especially if the “off-target” expression in GABAergic neurons is relatively sparse- if there is a small fraction of CA1 GABAergic interneurons expressing TeTX (but not the fluorophore, please see point 1 above) it is highly unlikely that even a relatively heroic amount of whole-cell recordings from randomly selected CA1 pyramidal cells would uncover a significant change in either mIPSCs or eIPSCs. Further, it is technically difficult to specifically stimulate specific GABAergic neurons (without labeling them via optogenetic channels) in the CaMKII-Cre: AAV.TeTX background. Importantly, when we designed the IPSC analysis in Figure 1, we specifically chose to record spontaneous IPSCs, as they include both spike-dependent and independent GABAergic transmission, in contrast to mIPSC, which are used to study GABAergic transmission in the absence of spikes. In short, the experiments suggested by the reviewer involve simultaneously expressing TeTX in CaMK2a::Cre+ neurons and optogenetic stimulation with ChR2 in interneurons followed by recording evoked IPSC in pyramidal cells – experiments that would require a substantial investment of both money and time, months to years, to basically look for a needle in a haystack, all in the face of other evidence to the contrary and previous work with these same reagents.
3. To illustrate the logic of our experiment design, at the request of the editor we have added a schematic to show how our perturbation impacts the local circuits in CA1 (Fig. 1r). In addition, in the discussion section of the revised manuscript discussion we now directly address the reviewer’s concerns in relation to the Veres et al paper (lines 292-300).

Reviewer #3 (Remarks to the Author):

The authors have thoroughly address my comments and I only have a minor remaining comment:

_ Regarding my comment on Fig.4d, I realized that the source of confusion came from the labeling of the color coded calibration bar (“High frequency power during HFEs (dB)”), which still need to be changed to something like “Pearson correlation coefficient” with a range of 0 to 1, as well as the figure legend (something like “Heatmaps show the correlation of high-frequency power between tetrodes during rest and exploration..” instead of “Heatmaps show the LFP power of high-frequency during ...”).

We thank the reviewer for their support and constructive comments. Per the reviewer’s suggestion, we have modified Fig.4d and the corresponding figure legend in the revised manuscript.

REVIEWERS' COMMENTS

Reviewer #2 (Remarks to the Author):

The most critical question still has not been addressed by the authors, namely the specificity of their intervention. They argue that the CaMKII promoter-driven circuit manipulation alters only pyramidal cell function. Indeed, the impairment of CA1 pyramidal cell output is nicely demonstrated in the manuscript. However, there is no direct evidence in this work ruling out the possibility that inhibitory cell function has not been impacted by the manipulations they applied. The key experiments need to be done, i.e. the direct functional assessment of GABAergic transmission, otherwise the main conclusion of the study does not stand on solid ground. If the intervention they used does not impact GABAergic circuit operation, then they should not see any change in inhibitory synaptic transmission, which would fully support the main conclusion of the study. The authors have clearly demonstrated their capability to record sIPSCs in CA1 pyramidal cells both in control and TeTx treated mice. Why the same experiments cannot be done in the presence of TTX to test the features of mIPSCs? They should see no change in the amplitude and rate of miniature events, if the CaMKII promoter driven manipulation impacted only pyramidal cells. Furthermore, the authors showed their capability to record eEPSCs in CA1 pyramidal cells. What is the reason that the same kind of experiment investigating the characteristics of IPSCs evoked by focal electrical stimulation or by light stimulation of the brain tissue expressing Chr2 under Dlx promoter in the two groups of mice cannot be achieved? These functional readouts (i.e., studying mIPSC and eIPSC characteristics) are critical control experiments. All other approaches provide only indirect evidence, making ambiguous the authors' conclusions. If the authors do not demonstrate directly that the GABAergic transmission in CA1 pyramidal cells functions normally upon TeTx expression in CA1 pyramidal cells, the conclusion of the present study has no firm support. Obviously, upon TeTx expression in CA1 pyramidal cells, a homeostatic compensation can occur in the hippocampal circuits, leading to downregulation in GABAergic transmission. In this case, the claim that feedback inhibition has a role in hippocampal oscillations is hard to justify as the feedforward inhibition may also be affected. If it turns out that there is a change in GABAergic transmission upon chronic silencing of CA1 pyramidal cell function, then the authors may overcome this limitation by applying acute silencing of CA1 pyramidal cell output and test the impact of this intervention on hippocampal oscillations.

Reviewer #3 (Remarks to the Author):

The authors adequately addressed all my comments.

Reviewer #2 (Remarks to the Author):

The most critical question still has not been addressed by the authors, namely the specificity of their intervention. They argue that the CaMKII promoter-driven circuit manipulation alters only pyramidal cell function. Indeed, the impairment of CA1 pyramidal cell output is nicely demonstrated in the manuscript. However, there is no direct evidence in this work ruling out the possibility that inhibitory cell function has not been impacted by the manipulations they applied. The key experiments need to be done, i.e. the direct functional assessment of GABAergic transmission, otherwise the main conclusion of the study does not stand on solid ground. If the intervention they used does not impact GABAergic circuit operation, then they should not see any change in inhibitory synaptic transmission, which would fully support the main conclusion of the study. The authors have clearly demonstrated their capability to record sIPSCs in CA1 pyramidal cells both in control and TeTx treated mice. Why the same experiments cannot be done in the presence of TTX to test the features of mIPSCs? They should see no change in the amplitude and rate of miniature events, if the CaMKII promoter driven manipulation impacted only pyramidal cells. Furthermore, the authors showed their capability to record eEPSCs in CA1 pyramidal cells. What is the reason that the same kind of experiment investigating the characteristics of IPSCs evoked by focal electrical stimulation or by light stimulation of the brain tissue expressing ChR2 under Dlx promoter in the two groups of mice cannot be achieved? These functional readouts (i.e., studying mIPSC and eIPSC characteristics) are critical control experiments. All other approaches provide only indirect evidence, making ambiguous the authors' conclusions. If the authors do not demonstrate directly that the GABAergic transmission in CA1 pyramidal cells functions normally upon TeTx expression in CA1 pyramidal cells, the conclusion of the present study has no firm support. Obviously, upon TeTx expression in CA1 pyramidal cells, a homeostatic compensation can occur in the hippocampal circuits, leading to downregulation in GABAergic transmission. In this case, the claim that feedback inhibition has a role in hippocampal oscillations is hard to justify as the feedforward inhibition may also be affected. If it turns out that there is a change in GABAergic transmission upon chronic silencing of CA1 pyramidal cell function, then the authors may overcome this limitation by applying acute silencing of CA1 pyramidal cell output and test the impact of this intervention on hippocampal oscillations.

We completely agree with the reviewer's assertion that the specificity of the genetic manipulation employed in these experiments is crucial to our interpretation of the data. However, as articulated in our previous revision, the reviewer suggested experiments involve enormous resources and time, and we believe they are not vital to substantiate the manuscript's central claims. Nonetheless, we present our detailed response below.

1. mIPSC: When we designed the IPSC analysis in Figure 1, we specifically chose to record spontaneous IPSCs, as they include both spike-dependent and independent GABAergic transmission, in contrast to mIPSC, which are used to study GABAergic transmission in the absence of spikes, and supplement with paired-pulse ratio analysis.
2. eIPSC: The reviewer previously cited Veres et al., paper and argued the nature of AAV.CaMKII-driven non-specific expression of the construct in inhibitory neurons (despite excitatory neuron-specific CaMKII promoter). We elaborated on the caveats of such manipulation in the manuscript discussion section (& to the reviewer in the response) and reiterated such an approach is vastly different from ours. Unfortunately, the reviewer is suggesting a complicated experiment again relying on the promoter, and this time, the Dlx promoter to precisely target

the interneurons in CaMKIIa:Cre mice background. We would argue that the reviewer's suggested experiment contradicts his/her views (between Veres et al., and ours; how do we know that AAV-driven Dlx-ChR2 will not express in excitatory neurons?). In addition, while we do have the capability, as the reviewer pointed out, the suggested experiment is expensive and laborious - it requires the authors to prepare a new AAV construct (Dlx-ChR2), optimize it, titer the viruses (Dlx-ChR2+ DIO-mCherry, or Dlx-ChR2+ DIO-TeTX.mCherry) for micro infusions, ex vivo recording, etc. for a revision. Moreover, interpretation of evoked IPSC in CA1 interneurons after electrical stimulation in DIO-TeTX.mCherry/ DIO-mCherry is confounded by many factors, including potential differences in interneuron subtypes recorded (as they are blind patching with mCherry- cells in CA1).

3. Acute silencing of outputs: Finally, as for the acute silencing experiment that the author suggested, we are unaware of any readily available method that allows us to block the pyramidal cell output acutely (while keeping the neurons electrically active, i.e., keeping the neurons generating action potentials); the unique ability of genetic TeTX expression to accomplish this was at the core of the design of these experiments. Nonetheless, we have added a line to the discussion noting that if possible this would be an ideal experiment.

Reviewer #3 (Remarks to the Author):

The authors adequately addressed all my comments.

We thank the reviewer for their support.